# How to Fine-tune the Model: Unified Model Shift and Model Bias Policy Optimization

**Hai Zhang  Hang Yu  Junqiao Zhao**\*  **Di Zhang**
**Chang Huang  Hongtu Zhou  Xiao Zhang  Chen Ye**
Department of Computer Science, Tongji University, Shanghai, China
MOE Key Lab of Embedded System and Service Computing, Tongji University, Shanghai, China
`{zhanghai12138, 2053881, zhaojunqiao}@tongji.edu.cn`

## Abstract

Designing and deriving effective model-based reinforcement learning (MBRL) algorithms with a performance improvement guarantee is challenging, mainly attributed to the high coupling between model learning and policy optimization. Many prior methods that rely on return discrepancy to guide model learning ignore the impacts of model shift, which can lead to performance deterioration due to excessive model updates. Other methods use performance difference bound to explicitly consider model shift. However, these methods rely on a fixed threshold to constrain model shift, resulting in a heavy dependence on the threshold and a lack of adaptability during the training process. In this paper, we theoretically derive an optimization objective that can unify model shift and model bias and then formulate a fine-tuning process. This process adaptively adjusts the model updates to get a performance improvement guarantee while avoiding model overfitting. Based on these, we develop a straightforward algorithm USB-PO[2] (Unified model Shift and model Bias Policy Optimization). Empirical results show that USB-PO achieves state-of-the-art performance on several challenging benchmark tasks.

## 1 Introduction

Nowadays, reinforcement learning (RL) has been gaining much traction in a wide variety of complicated decision-making tasks ranging from academia to industry [42, 9, 5, 13, 28]. Part of this is due to some remarkable model-free RL (MFRL) algorithms [40, 15, 11, 41, 18], which show desirable asymptotic performance. However, their applications are hindered by the bottleneck of sample efficiency. On the contrary, model-based RL (MBRL) algorithms, using a world model to generate the imaginary rollouts and then taking them for policy optimization [31, 20], have high sample efficiency while achieving similar asymptotic performance, thus becoming a compelling alternative in practical cases [39, 16, 52].

Typically, MBRL algorithms iterate between model learning and policy optimization. Hence the model quality is crucial for MBRL. Many prior methods [31, 20, 53, 38, 24] rely on return discrepancy to obtain model updates with a performance improvement guarantee. While having achieved comparable results, they only account for model bias in one iteration [20] but do not consider the impacts of model shift between two iterations [21], which can lead to performance deterioration due to excessive model updates. Although CMLO [21] explicitly considers model shift from the perspective of the performance difference bound, it only sets a fixed threshold to constrain the impacts of model shift and determines when the model should be updated accordingly. If this threshold is

---

\*Corresponding author
[2]Code: https://github.com/betray12138/Unified-Model-Shift-and-Model-Bias-Policy-Optimization.git

37th Conference on Neural Information Processing Systems (NeurIPS 2023).

set too low, the model bias of the following iteration will be large, which impairs the subsequent optimization process. If this threshold is set too high, the performance improvement can no longer be guaranteed. We remark that such an update is heavily dependent on the choice of this threshold and should be adjusted adaptively during the training process. Therefore, a smarter scheme is required to unify model shift and model bias, enabling adaptively adjusting their impacts to get a performance improvement guarantee.

In this paper, we theoretically derive an optimization objective that can unify model shift and model bias. Specifically, according to the performance difference bound, we propose to minimize the sum of two terms: the model shift term between the pre-update model and the post-update model and the model bias term of the post-update model, each of which is denoted by a second-order Wasserstein distance [47]. By minimizing this optimization objective after the model update via maximum likelihood estimation (MLE) [6, 20], we can tune the model to adaptively find appropriate updates to get a performance improvement guarantee.

Based on these, we develop a straightforward algorithm USB-PO (Unified model Shift and model Bias Policy Optimization). To the best of our knowledge, this is the first method that unifies model shift and model bias and adaptively fine-tunes the model updates during the training process. We evaluate USB-PO on several continuous control benchmark tasks. The results show that USB-PO has higher sample efficiency and better final performance than other state-of-the-art (SOTA) MBRL methods and yields promising asymptotic performance compared with the MFRL counterparts.

## 2   Related works

MBRL algorithms are promising candidates for real-world sequential decision-making problems due to their high sample efficiency. Existing studies can be divided into several categories [4, 45, 1, 34, 49, 8, 17] following their different usage of the model. Our work falls into the Dyna-style category [45, 44]. Specifically, after model learning, the model generates the imaginary rollouts into the replay buffer for subsequent policy optimization. Hence, both model learning and policy optimization have critical impacts on asymptotic performance.

Some previous algorithms focus on the policy optimization process. CMBAC [51] introduces conservatism to reduce overestimation of the action value function, and ME-TRPO [23] imposes constraints on the policy to get reliable updates within the trust region. Our work is oriented towards model learning, which is orthogonal to these algorithms. Hence, we are inclined to propose a generic algorithm similar to [24, 20, 21] that can be plugged into many SOTA MFRL algorithms [15, 29], rather than just proposing for a specific policy optimization algorithm.

A key issue in model learning is model bias, which refers to the error between the model and the real environment [20]. As the imaginary rollout horizon increases, the impacts of model bias accumulate rapidly, leading to compounding error and unreliable transitions. To mitigate this problem, several effective methods have been proposed. The ensemble model technique [6, 25, 37] and the dropout method [12] are employed to prevent model overfitting. The uncertainty estimation techniques are used to adjust the rollout length [20, 30] or the transition weight [19, 38]. Furthermore, The multi-step techniques [2, 48] are applied to prevent direct input of the imaginary states. We follow the previous work [20] to use the combination of the ensemble model technique with short model rollouts to mitigate the compounding error.

Performance improvement guarantee is a core concern in both MFRL and MBRL theoretic avenues. In MFRL, methods such as TRPO [41] and CPI [22] choose to optimize the performance difference bound, whilst most of the previous work in MBRL [31, 20, 53, 38, 24] choose to optimize the difference of expected return under the model and that of the real environment, which is termed return discrepancy. However, return discrepancy ignores model shift between two consecutive iterations compared to the performance difference bound under the MBRL setting, which can lead to performance deterioration due to excessive model updates. Although some recent methods have also employed performance difference bound to construct theoretical proofs, they still suffer from certain limitations. OPC [10] designs an algorithm to optimize on-policy model error, but it is similar to return discrepancy in nature. DPI [43] uses dual updates to improve sample efficiency but tries to restrict policy updates within the trust region, thus inhibiting exploration. CMLO [21] relies on a fixed threshold to constrain the impacts of model shift, resulting in a heavy dependence on the threshold and a lack of adaptability during the training process. Hence, we try to unify model shift

and model bias to form a novel optimization problem, adaptively fine-tuning the model updates to get a performance improvement guarantee. Still, some prior work [7, 36, 54] choose to consider regret bound, among which [54] also reduce the impacts of the model changing dramatically between successive iterations. Instead of unifying model shift and model bias, they choose to realize dual optimization by considering maximizing the expectation of the model value rather than that of the single model as a sub-process. Different from [54, 43, 27] that use dual optimization to train the policy, we devise an extra phase to fine-tune the model.

## 3   Preliminaries

We consider a Markov Decision Process (MDP), defined by the tuple $M = (\mathcal{S}, \mathcal{A}, p, r, \gamma, \rho_0)$. $\mathcal{S}$ and $\mathcal{A}$ denote the state space and action space respectively, and $\gamma \in (0, 1)$ denotes the discount factor. $p(s'|s, a)$ denotes the dynamic transition distribution and we denote $p_{M^*}(s'|s, a)$ as that of the real environment. $\rho_0(s)$ denotes the initial state distribution and $r(s, a)$ denotes the reward function. RL aims to find the optimal policy $\pi^*$ that maximizes the expected return under the real environment $M^*$ denoted by the value function $V_{M^*}^\pi$ as Eq.(1):

$$\pi^* = \arg\max_\pi V_{M^*}^\pi = \mathbb{E}_{a_t \sim \pi(\cdot|s_t), s_{t+1} \sim p_{M^*}(\cdot|s_t, a_t)}[\sum_{t=0}^\infty \gamma^t r(s_t, a_t)|\pi, s_0], s_0 \sim \rho_0(s) \qquad (1)$$

MBRL algorithms aim to learn the dynamic transition distribution model, $p_M(s'|s, a)$, by using samples collected from interaction with the real environment via supervised learning. We denote the expected return under the model $M$ of the policy $\pi$ as $V_M^\pi$ and denote that under the real environment of the policy $\pi$ derived from the model $M$ as $V^{\pi|M}$. Additionally, we assume that $r(s, a)$ is unknown to the model $M$ and the model will predict $r_M$ as the reward function. Besides, we denote $\mathcal{M}$ as a parameterized family of models and $\Pi$ as a parameterized family of policies.

Let $d_M^\pi(s, a)$ denote the normalized discounted visitation probability for $(s, a)$ when starting at $s_0 \sim \rho_0$ and following $\pi$ under the model $M$. Let $p_{M,t}^\pi(s)$ denote the probability of visiting $s$ at timestep $t$ given the policy $\pi$ and the model $M$.

$$d_M^\pi(s, a) = (1 - \gamma) \sum_{t=0}^\infty \gamma^t p_{M,t}^\pi(s) \pi(a|s) \qquad (2)$$

We define the total variation distance (TVD) estimator as $D_{TV}(\cdot||\cdot)$ and the second-order Wasserstein distance estimator as $W_2(\cdot, \cdot)$.

## 4   USB-PO framework

In this section, we demonstrate a detailed description of our proposed algorithmic framework. i.e., USB-PO. In Section 4.1, a meta-algorithm of the USB-PO framework is provided as a generic solution. In Section 4.2, we theoretically show how to unify model shift and model bias to get a performance improvement guarantee[3]. In Section 4.3, the practical algorithm is proposed to instantiate the USB-PO framework.

### 4.1   The overall algorithm

The general algorithmic framework of USB-PO is depicted in Algorithm 1, where the main difference compared to the existing MBRL algorithms is the two-phase model learning process, namely phase 1 and phase 2. Phase 1 uses traditional MLE loss to train the model, which may impair the performance by excessive model updates due to only considering the impacts of model bias. To mitigate this problem, we introduce phase 2 to further fine-tune the model updates, whose optimization objective is defined as Eq.(3).

$$\arg\min_{p_{M_2}} \mathcal{J}_{phase2} = \mathbb{E}_{d_{M_1}^\pi}[W_2(p_{M_1}, p_{M_2}) + W_2(p_{M_2}, p_{M^*})] \qquad (3)$$

---

[3]The detailed derivations of all the theorems in this section are presented in the appendix.

**Algorithm 1** Meta-Algorithm of the USB-PO Framework
___
1: Initialize the policy $\pi$ and the learned model;
2: Initialize the environment replay buffer $\mathcal{D}$ and the model replay buffer $\mathcal{D}_M$;
3: **for** each epoch **do**
4:     Use $\pi$ to interact with the real environment: $\mathcal{D} \leftarrow \mathcal{D} \cup \{(s, a, r, s')\}$;
5:     Backup the current learned model for future use and denote this backed-up model as $p_{M_1}$;
6:     Phase 1: use $\mathcal{D}$ to train the learned model with the supervision of MLE and denote this updated model as $p_{M_2}$;
7:     Phase 2: use Eq.(3) as optimization objective to further fine-tune $p_{M_2}$;
8:     Use $p_{M_2}$ to generate the imaginary rollouts: $\mathcal{D}_M \leftarrow \mathcal{D}_M \cup \{(s_M, a_M, r_M, s'_M)\}$;
9:     Use $\mathcal{D} \cup \mathcal{D}_M$ to train the policy $\pi$
10: **end for**
___

Eq.(3) unifies the model shift term and the model bias term in the second-order Wasserstein distance form, namely $W_2(p_{M_1}, p_{M_2})$ and $W_2(p_{M_2}, p_{M^*})$, thus achieving adaptive adjustment of their impacts during the fine-tuning process. As demonstrated in Section 4.2 and Section 5.4, this is not equivalent to the traditional methods of limiting the magnitude of model updates, but rather beneficial to get a performance improvement guarantee.

## 4.2 Theoretical proof

**Definition 1** (Performance Difference Bound). *Recalling that $V^{\pi_i|M_i}$ denotes the expected return under the real environment of the policy $\pi_i \in \Pi$ derived from the model $M_i \in \mathcal{M}$. The lower bound on the true return gap of $\pi_1$ and $\pi_2$ can be stated as,*

$$V^{\pi_2|M_2} - V^{\pi_1|M_1} \geq C \tag{4}$$

By constantly increasing the value of $C$, the lower bound on the performance difference is guaranteed to be lifted, leading to performance improvement. Therefore, we try to take model shift and model bias into the formulation of $C$ and maximize it to achieve our goal.

**Theorem 1** (Performance Difference Bound Decomposition). *Let $M_i \in \mathcal{M}$ be the evaluated model and $\pi_i \in \Pi$ be the policy derived from the model. The performance difference bound can be decomposed into three terms,*

$$V^{\pi_2|M_2} - V^{\pi_1|M_1} = (V^{\pi_2|M_2} - V_{M_2}^{\pi_2}) - (V^{\pi_1|M_1} - V_{M_1}^{\pi_1}) + (V_{M_2}^{\pi_2} - V_{M_1}^{\pi_1}) \tag{5}$$

Obviously, compared to directly optimizing the return discrepancy of each iteration [20], the performance difference bound chooses to optimize the return discrepancy of two adjacent iterations, namely $V^{\pi_2|M_2} - V_{M_2}^{\pi_2}$ and $V^{\pi_1|M_1} - V_{M_1}^{\pi_1}$ respectively, and the expected return variation between these two iterations, namely $V_{M_2}^{\pi_2} - V_{M_1}^{\pi_1}$, demonstrating better rigorousness. For further discussion, we introduce the following theorem.

**Theorem 2** (Return Bound). *Let $R_{max}$ denote the bound of the reward function, $\epsilon_\pi$ denote $\max_s D_{TV}(\pi_1 || \pi_2)$ and $\epsilon_{M_1}^{M_2}$ denote $\mathbb{E}_{(s,a) \sim d_{M_1}^{\pi_1}}[D_{TV}(p_{M_1} || p_{M_2})]$. For two arbitrary policies $\pi_1, \pi_2 \in \Pi$, the expected return under two arbitrary models $M_1, M_2 \in \mathcal{M}$ can be bounded as,*

$$V_{M_2}^{\pi_2} - V_{M_1}^{\pi_1} \geq -2R_{max}\left(\frac{\epsilon_\pi}{(1-\gamma)^2} + \frac{\gamma}{(1-\gamma)^2}\epsilon_{M_1}^{M_2}\right) \tag{6}$$

By using Eq.(6), we can easily bound the decomposition terms of the performance difference bound in Eq.(5).

**Theorem 3** (Decomposition TVD Bound). *Let $\epsilon_{M_i}^{\pi_i}$ denote $\mathbb{E}_{(s,a) \sim d_{M_i}^{\pi_i}}[D_{TV}(p_{M_i} || p_{M^*})]$. Let $M_i \in \mathcal{M}$ be the evaluated model and $\pi_i \in \Pi$ be the policy derived from the model. The decomposition terms can be bounded as,*

$$V^{\pi_2|M_2} - V^{\pi_1|M_1} \geq \frac{2R_{max}\gamma}{(1-\gamma)^2}(\epsilon_{M_1}^{\pi_1} - \epsilon_{M_2}^{\pi_2} - \epsilon_{M_1}^{M_2}) - \frac{2R_{max}\epsilon_\pi}{(1-\gamma)^2} \tag{7}$$

Notably, the model shift term and the model bias term in TVD form, namely $D_{TV}(p_{M_1}||p_{M_2})$ and $D_{TV}(p_{M_2}||p_{M^*})$, are already present in the right-hand side of the Eq.(7). However, due to the unknown term $\pi_2$, we can not sample from $d_{M_2}^{\pi_2}$. Thus, we need to make a further transformation about the Eq.(7).

**Theorem 4** (Unified Model Shift and Model Bias Bound). *Let $\kappa$ denote the constant $\frac{2R_{max}}{(1-\gamma)^2}$ and $\Delta$ denotes $\mathbb{E}_{(s,a)\sim d_{M_1}^{\pi_1}}[D_{TV}(p_{M_2}||p_{M^*})] - \mathbb{E}_{(s,a)\sim d_{M_2}^{\pi_2}}[D_{TV}(p_{M_2}||p_{M^*})]$. Let $M_i \in \mathcal{M}$ be the evaluated model and $\pi_i \in \Pi$ be the policy derived from the model. The unified model shift and model bias bound can be derived as,*

$$
\begin{aligned}
&V^{\pi_2|M_2} - V^{\pi_1|M_1} \\
&\geq \kappa(\gamma(\mathbb{E}_{(s,a)\sim d_{M_1}^{\pi_1}}[D_{TV}(p_{M_1}||p_{M^*}) - D_{TV}(p_{M_1}||p_{M_2}) - D_{TV}(p_{M_2}||p_{M_*})] + \Delta) - \epsilon_\pi)
\end{aligned} \tag{8}
$$

The $\Delta$ term in Eq.(8) is still intractable. However, the fact that it covers $d_{M_1}^{\pi_1}$ and $d_{M_2}^{\pi_2}$ reminds us that we may explore its relationship with the three TVD terms lying in the expectation of $d_{M_1}^{\pi_1}$.

**Theorem 5** (|$\Delta$| Upper Bound). *Let $M_i \in \mathcal{M}$ be the evaluated model and $\pi_i \in \Pi$ be the policy derived from the model. The term $\Delta$ can be upper bounded as:*

$$
|\Delta| \leq \frac{2\gamma}{1-\gamma}\mathbb{E}_{(s,a)\sim d_{M_1}^{\pi_1}}[D_{TV}(p_{M_1}||p_{M_2})\max_{s,a} D_{TV}(p_{M_2}||p_{M^*})] + \frac{2\epsilon_\pi}{1-\gamma}\max_{s,a} D_{TV}(p_{M_2}||p_{M^*}) \tag{9}
$$

Under the online setting, it is assumed that the error caused by policy shift, namely $\epsilon_\pi$, compared to model bias has a relatively small scale [20, 53]. Therefore, we ignore the minor influence brought by the policy. Additionally, the terms lying in the expectation of $d_{M_1}^{\pi_1}$ in Eq.(9) is the product of the model shift term and the model bias term in TVD form, both of which have the range $[0, 1]$ [26], making the upper bound of |$\Delta$| become a higher-order term compared to each of them alone. Thus, the critical element to lift the lower bound in Eq.(8) is the terms lying in the expectation of $d_{M_1}^{\pi_1}$. By maximizing these terms, namely minimizing $D_{TV}(p_{M_1}||p_{M_2}) + D_{TV}(p_{M_2}||p_{M^*})^4$, we can achieve unifying model shift and model bias and adaptively fine-tuning the model updates to get a performance improvement guarantee.

To make it practically feasible, we make additional assumptions. Specifically, let $(\mathcal{X}, \Sigma)$ be a measurable space and $\mathcal{F}$ is a set of functions mapping $\mathcal{X}$ to $\mathbb{R}$ that contains $V_M^\pi$. When $V_M^\pi$ is $L_v$-Lipschitz with respect to a norm $||\cdot||$, the integral probability metric [33] of two arbitrary dynamic transition function $M, M' \in \mathcal{M}$ defined on $\mathcal{X}$ is as Eq.(10). Notice that if $1 \leq p \leq q$, $W_p(p_M, p_{M'}) \leq W_q(p_M, p_{M'})$ [32], and we can get $W_1(p_M, p_{M'}) \leq W_2(p_M, p_{M'})$. Hence, Eq.(3) can be applied as the optimization objective to get a performance improvement guarantee.

$$
\sup_{f\in\mathcal{F}}|\mathbb{E}_{s'\sim p_M}[f(s')] - \mathbb{E}_{s'\sim p_{M'}}[f(s')]| = \frac{R_{max}}{1-\gamma}D_{TV}(p_M||p_{M'}) = L_v W_1(p_M, p_{M'}) \tag{10}
$$

### 4.3 Practical implementation

We now instantiate Algorithm 1 by demonstrating an explicit approach. To better clarify, we would like to state the four design decisions in detail: (1) how to parametrize the model, (2) how to estimate the model bias term and model shift term in phase 2, (3) how to use the model to generate rollouts and (4) how to use the model rollouts to optimize the policy $\pi$.

**Predictive Model.** We use a bootstrap ensemble of dynamic models $\{p_M^1, ..., p_M^B\}$, whose elements are all probabilistic neural network, outputting the Gaussian distribution with diagonal covariance: $p_M^i(s_{t+1}, r_{t+1}|s_t, a_t) = \mathcal{N}(\mu_M^i(s_t, a_t), \Sigma_M^i(s_t, a_t))$. The probabilistic ensemble model can capture the aleatoric uncertainty arising from inherent stochasticities and the epistemic uncertainty corresponding to ambiguously determining the underlying system due to the lack of sufficient data [6]. In the field of MBRL, properly handling these uncertainties can achieve better asymptotical performance. Following the previous work [20], we select a model uniformly from the elite set to generate the transition at each step in the rollout process. In phase 1, the ensemble models are trained on shared but differently shuffled data, where the optimization objective is MLE [6, 20].

---

[4]$D_{TV}(p_{M_1}||p_{M^*})$ is a constant for the optimization of $M_2$

**Model Shift and Model Bias Estimation.** Recalling that the model shift term refers to $W_2(p_{M_1}||p_{M_2})$ and the model bias term refers to $W_2(p_{M_2}||p_{M^*})$. For arbitrary two Gaussian distributions $p \sim \mathcal{N}(\mu_1, \Sigma_1)$ and $q \sim \mathcal{N}(\mu_2, \Sigma_2)$, the second-order Wasserstein distance between them is derived as Eq.(11) [35].

$$W_2(p, q) = \sqrt{||\mu_1 - \mu_2||_2^2 + \text{trace}(\Sigma_1 + \Sigma_2 - 2(\Sigma_2^{\frac{1}{2}}\Sigma_1\Sigma_2^{\frac{1}{2}})^{\frac{1}{2}})} \tag{11}$$

Let $k_1$ denote the index of the selected model from $M_1$ ensemble and $k_2$ denote that from $M_2$ ensemble. For the model shift term, we approximate it as $W_2(p_{M_1}^{k_1}, p_{M_2}^{k_2})$. For the model bias term, we approximate it as $\frac{1}{B-1} \sum_{b=1, b \neq k_2}^{B} W_2(p_{M_2}^{k_2}, p_{M_2}^b)$.

**Model Rollout.** Following the previous work [20], we use short branch rollouts to alleviate the impacts of compounding error.

**Policy Optimization.** Our work allows being plugged in many SOTA MFRL algorithms, e.g. SAC [15], TD3 [11], etc. Here we employ SAC as an example.

## 5 Experiment

Our experiments are designed to investigate four primary questions: (1) Whether we need to propose an algorithm similar to USB-PO to adaptively adjust the impacts of model shift? (2) How well does USB-PO perform on reinforcement learning benchmark tasks compared to SOTA MBRL and MFRL algorithms? (3) How to understand USB-PO? (4) Does USB-PO have a similar performance on different learning rate settings in phase 2?

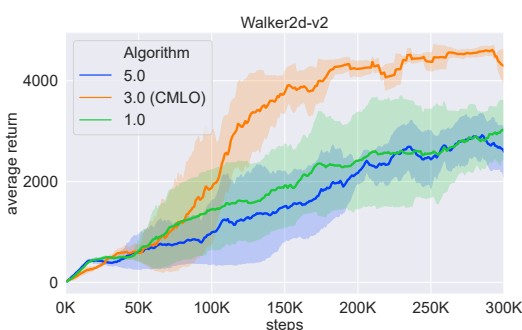

### 5.1 Necessity of USB-PO

We recall that CMLO [21] sets a fixed threshold to constrain the impacts of model shift, which we argue CMLO is heavily dependent on this threshold. To valid our statement, we devise an experiment that sets three different thresholds for CMLO on Walker2d environment in MuJoCo [46]. As Figure 1 shows, the performance corresponding to the other two thresholds (1.0 and 5.0) is severely affected. Therefore, setting a fixed threshold to constrain is inappropriate, and a smarter way like USB-PO should be applied to adaptively adjust the impacts of model shift.

Figure 1: CMLO performance curves for different threshold settings over different random seeds, where 3.0 is the threshold recommended in the paper.

### 5.2 Comparison with baselines

To highlight our algorithm, we compare USB-PO against some SOTA MBRL and MFRL algorithms. The MBRL baselines include CMLO [21], which carefully chooses the threshold for different environments to constrain the impacts of model shift, ALM [14], which learns the representation, model, and policy into one objective, PDML [50], which uses historical policy sequence to aim the model training process, MBPO [20], which is the variant without our proposed model fine-tuning process and uses return discrepancy to get a performance improvement guarantee. The MFRL baselines include SAC [15], the SOTA MFRL algorithm in terms of asymptotic performance and PPO [40], which explores monotonic improvement under the model-free setting.

We evaluate USB-PO and these baselines on six MuJoCo [46] continuous control tasks in OpenAI Gym [3], covering Humanoid, Walker2d, Ant, HalfCheetah, Hopper and Inverted-Pendulum, with more details showing in the appendix. To be fair, we employ the standard 1000-step version of these tasks with the same environment settings.

Figure 2 shows the learning curves of all compared methods, together with the asymptotic performance. The results show that our algorithm is remarkably advanced over the MFRL algorithms

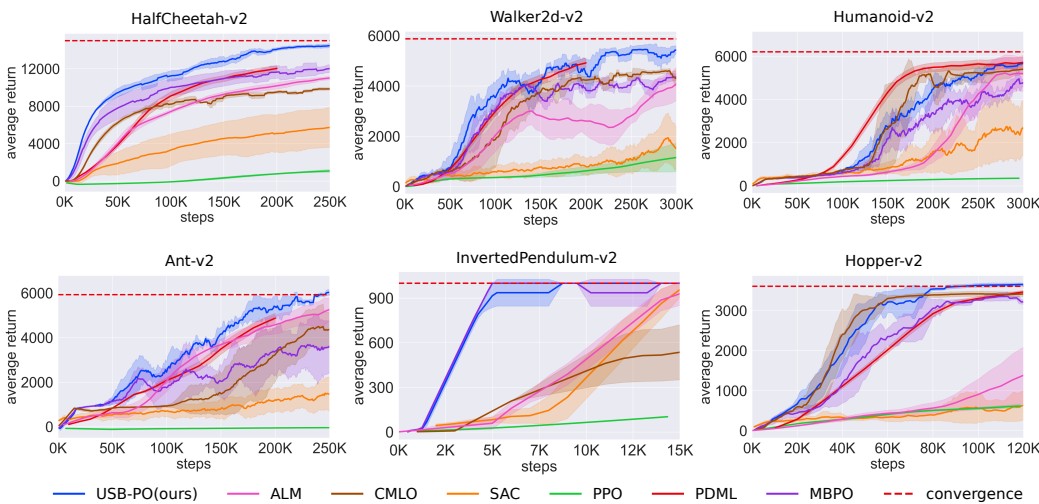

Figure 2: Comparison against baselines on continuous control benchmarks. Solid curves refer to the mean performance of trials over different random seeds, and shaded area refers to the standard deviation of these trials. Dashed lines refer to the asymptotic performance of SAC (at 3M steps). Note that PDML is not evaluated in InvertedPendulum-v2.

regarding sample efficiency, along with asymptotic performance on par with the SOTA MFRL algorithm SAC. Compared to the MBRL baselines, our method achieves higher sample efficiency and better final performance. Notably, compared to CMLO, which requires a finely chosen threshold for each environment to constrain the impacts of model shift, our method utilizes the same learning rate for the fine-tuning process in all environments. This further validates unifying the model shift and the model bias to adjust the model updates adaptively is reasonable.

**Computational Cost.** We report our computational cost compared to MBPO [20] in Appendix D.4. Although USB-PO is a two-phase model training process, continuing to use the fine-tuned model for the next iteration has the potential to accelerate model convergence and then possibly reduce the training time.

### 5.3 How to understand USB-PO

In this section, we first design an experiment to illustrate the value magnitude of $\Delta$, the model shift term and the model bias term, validating whether using the second-order Wasserstein distance satisfies the prerequisites for getting a performance improvement guarantee, and then use more in-depth experiments to illustrate how USB-PO works and show its superiority.

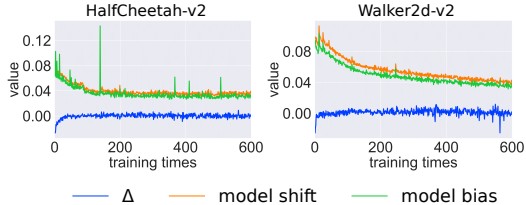

Figure 3: The value magnitude of the $\Delta$, the model shift term and the model bias term during the training process in HalfCheetah and Walker2d.

**Value Magnitude.** We choose 2 challenging tasks in MuJoCo, HalfCheetah and Walker2d, and plot the value of $\Delta$, the model shift term and the model bias term during the training process respectively. To approximate the $\Delta$ practically, we use the samples from the model replay buffer before the update of the model and the policy to approximate sampling from $d_{M_1}^{\pi_1}$ and use the samples generated by them after update to approximate sampling from $d_{M_2}^{\pi_2}$.

As shown in Figure 3, the magnitude of $\Delta$ generally oscillates in a slight manner around 0 while the magnitude of model shift term and the model bias term are much larger than $\Delta$ even in the converged cases. Therefore, it is reasonable to ignore the impacts of $\Delta$ in Eq.(8). This further supports that minimizing Eq.(3) can get a performance improvement guarantee.

**Working Mechanism.** To illustrate the significance of our method, we calculate the optimization objective value before and after the fine-tuning process and plot their difference, as shown in Figure 4. All random seeds show the consistent result that at the beginning of training the sum of model bias and model shift is large and thus the fine-tuning magnitude is relatively large. As the model is trained to converge, the fine-tuning process also gradually converges to ensure the stability of the model. Notice that in some cases the fine-tuning process does not choose to fine-tune the model updates actually (0 refers to no update, which means the updates in phase 1 are reasonable), which further supports our theorem and the motivation of this paper, i.e., actively adaptive adjustment of the model updates to get a performance improvement guarantee, rather than passively waiting for model shift to surpass a given threshold to make a substantial update [21]. Based on the performance comparison in Figure 2, we argue that actively adaptive updates are more beneficial.

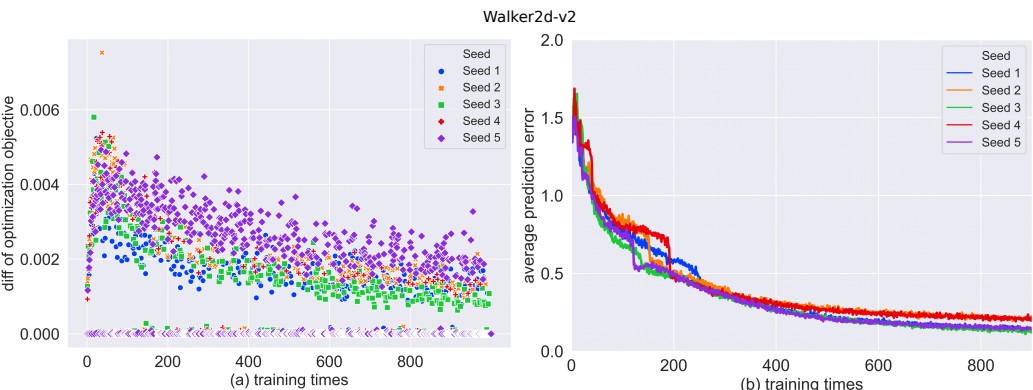

Figure 4: (a) the difference of the optimization objective value before and after the fine-tuning process during the training process over 5 random seeds. (b) the average prediction error during the training process over these random seeds. The model is generally near convergence when training times reach 1K.

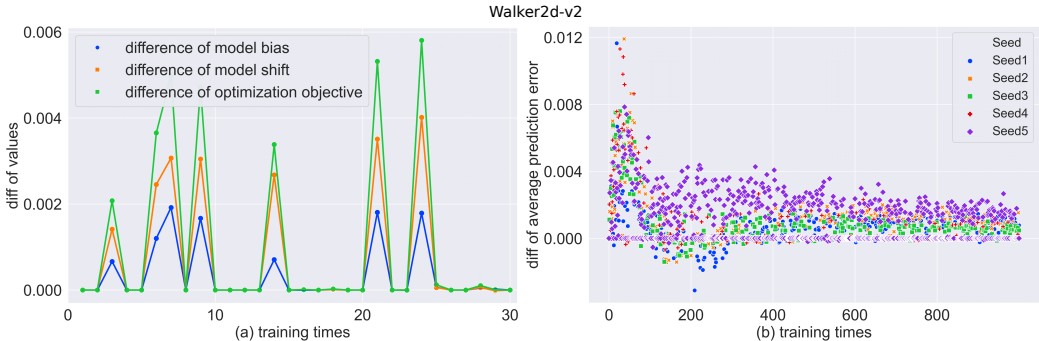

Figure 5: (a) we choose a specific random seed to show the details of the first 30 training times, covering the difference of optimization objective value, the model shift term and the model bias term before and after the fine-tuning process. (b) the difference of average prediction error before and after the fine-tuning process during the training process over the previously used random seeds.

We further plot the difference of the model shift term, the model bias term and the average prediction error before and after the fine-tuning process. As shown in Figure 5 (a), when the fine-tuning actually operates, the difference of the model shift term and the model bias term are both positive. As shown in Figure 5 (b), the fine-tuning process has a positive effect on the reduction of the average prediction error. These observations suggest that USB-PO has other positive effects during the fine-tuning process, i.e. potentially reducing both model shift and model bias and leading to model overfitting avoidance.

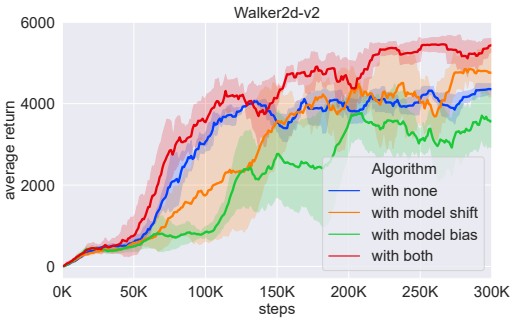 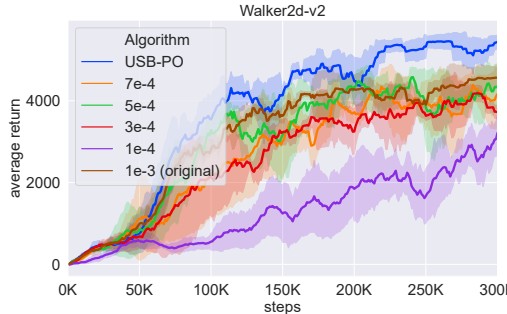

Figure 6: The average return of different optimization objective variants over different random seeds.

Figure 7: The average return of USB-PO and MBPO with different learning rates over different random seeds.

Additionally, the change in the adjustment magnitude of the average prediction error indicates that the model can converge under the role of the fine-tuning process, which is consistent with Figure 4, forming bi-verification.

To conclude, when USB-PO actively recognizes that an improper update happens, it performs a pull-back operation (the model fine-tuning process) to secure the performance improvement guarantee while avoiding model overfitting. In contrast, when USB-PO considers that a reasonable update is done, no actual operation will be taken. According to the performance comparison results in Figure 2, it is verified that the model quality plays a determinant role in the Dyna-Q [44] algorithms.

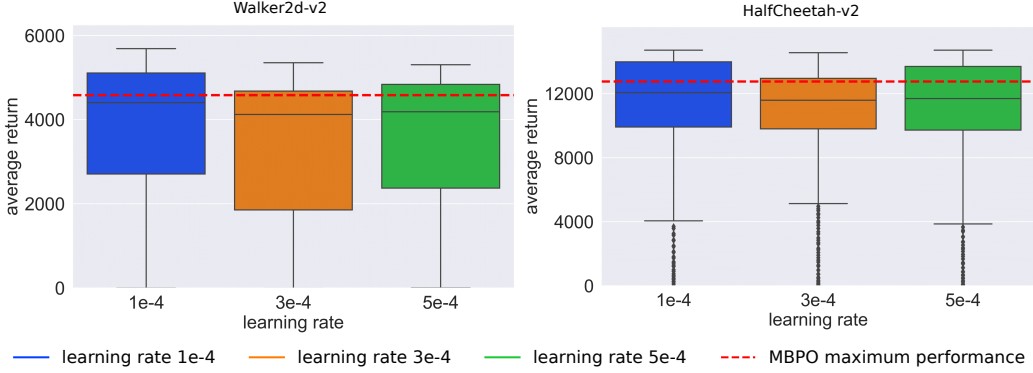

Figure 8: Performance of different learning rates on Walker2d and HalfCheetah during the whole training process. Each learning rate is repeated several times over different random seeds and the dashed line refers to the maximum average returns of MBPO.

## 5.4 Ablation study

In this section, we design 3 ablation experiments to strengthen our superiority.

**Optimization Objective Variants.** We set three variants for our optimization objective, covering (1) without the model shift term and the model bias term, which is equal to MBPO [20], (2) with the model shift term, (3) with the model bias term. As shown in Figure 6, only optimizing the model shift term results in a drop in sample efficiency since the fine-tuning process makes $M_2$ tend to update towards $M_1$ and only optimizing the model bias term leads to performance deterioration due to excessive model updates.

**Not Equivalent to Limiting the Update Magnitude.** We set different learning rates for MBPO to compare with USB-PO. As Figure 7 shows, USB-PO is more dominant from the perspective of the average return, which further strengthens that USB-PO is not equivalent to limiting the update magnitude, but rather beneficial to get a performance improvement guarantee.

**Learning Rate Performance Comparison.**    We set up an ablation experiment on the learning rate of the fine-tuning process. As shown in Figure 8, unlike CMLO which is strongly dependent on a carefully chosen threshold for each environment to constrain the impacts of model shift, USB-PO is less sensitive to the learning rate of phase 2.

## 6    Discussion

In this paper, we propose a novel MBRL algorithm called USB-PO, which can unify model shift and model bias, enabling adaptive adjustment of their impacts to get a performance improvement guarantee. We further find that our method can potentially reduce both model shift and model bias, leading to model overfitting avoidance. Empirical results on several challenging benchmark tasks validate the superiority of our algorithm and more in-depth experiments are conducted to demonstrate our mechanism. The limitation of our work is we do not investigate the change of model shift and model bias in this optimization problem theoretically. Therefore, one direction that merits further research is how to solidly interpret the variation of model shift and model bias in this fine-tuning process. Further, we aim to propose a general algorithmic framework in this paper. Due to the simple design of our model architecture, we find it difficult to capture some complex and practical locomotion. We hope to combine our adaptive adjusting technique with more advanced model architectures in the future.

## Acknowledgments and Disclosure of Funding

This work is supported by the National Key Research and Development Program of China (No. 2021YFB2501104 No. 2020YFA0711402). We thank Michael Janner, Raj Ghugare, Zifan Wu, and Xiyao Wang for providing the baseline results and thank Tianying Ji for the worthy discussions.

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

# 7 Appendix

## A Proof Sketch

To better clarify our theoretical results, we provide a proof sketch here. Firstly, we decompose the performance difference bound under the model-based setting into three terms (Theorem 1). Secondly, by means of using Return Bound (Theorem 2), we can bound these three terms individually (Theorem 3). Then, we can do some transformation to get Unified Model Shift and Model Bias Bound (Theorem 4), which bounds the model shift term and the model bias term in total variation form. However, due to the intractable property of $\Delta$, we further explore the upper bound of $|\Delta|$ (Theorem 5), finding that $\Delta$ can be ignored. Finally, by the Integral Probability Metrics (Lemma 3) and the property of the Wasserstein distance, we derive the target which bounds the model shift term and the model bias term in the Wasserstein distance form.

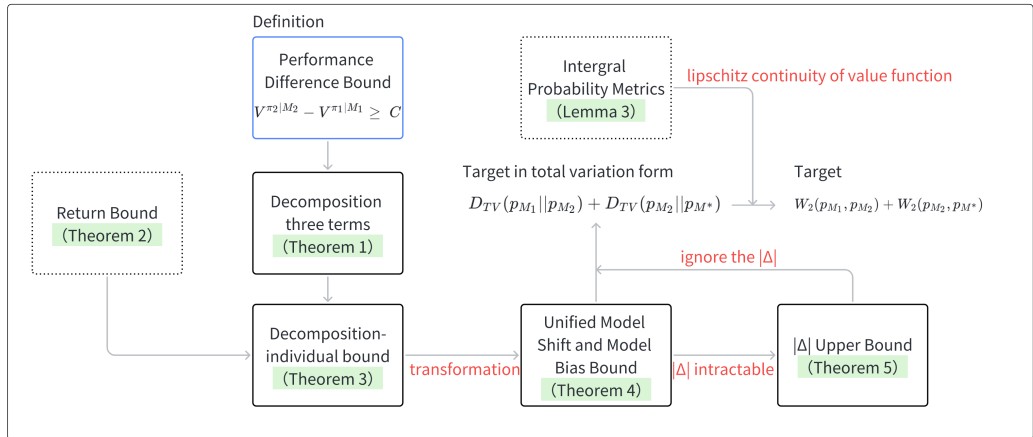

Figure 9: Theoretical sketch of USB-PO.

## B Useful Lemmas

In this section, we provide some proof to support our theoretical analysis.

**Lemma 1** (Total variation Distance). *Consider a measurable space* $(\Omega, \Sigma)$ *and probability measures* $P$ *and* $Q$ *are defined on* $(\Omega, \Sigma)$. *The total variation distance between* $P$ *and* $Q$ *is defined as:*

$$D_{TV}(P||Q) = \sup_{A \in \Sigma} |P(A) - Q(A)| \tag{12}$$

*Eq.(12) can be equivalently written as:*

$$D_{TV}(P||Q) = \frac{1}{2} \sum_{\omega \in \Omega} |P(\{\omega\}) - Q(\{\omega\})| \tag{13}$$

*Proof:* The proof of this lemma can be found in [26]. □

**Lemma 2** (Total Variation Distance of Joint Distributions). *Given two distributions* $p(x, y) = p(x)p(y|x)$ *and* $q(x, y) = q(x)q(y|x)$, *the total variation distance between them can be bounded as:*

$$D_{TV}(p(x, y)||q(x, y)) \leq D_{TV}(p(x)||q(x)) + \max_{x} D_{TV}(p(y|x)||q(y|x)) \tag{14}$$

*Proof:*

$$D_{TV}(p(x,y)||q(x,y)) = \frac{1}{2}\sum_{x,y}|p(x,y) - q(x,y)|$$

$$= \frac{1}{2}\sum_{x,y}|p(x)p(y|x) - q(x)q(y|x)|$$

$$= \frac{1}{2}\sum_{x,y}|p(x)p(y|x) - p(x)q(y|x) + p(x)q(y|x) - q(x)q(y|x)|$$

$$\leq \frac{1}{2}\sum_{x,y}p(x)|p(y|x) - q(y|x)| + |p(x) - q(x)|q(y|x)$$

$$= \frac{1}{2}\sum_{x,y}p(x)|p(y|x) - q(y|x)| + \frac{1}{2}\sum_{x}|p(x) - q(x)|$$

$$= \mathbb{E}_{x\sim p(x)}[D_{TV}(p(y|x)||q(y|x))] + D_{TV}(p(x)||q(x))$$

$$\leq D_{TV}(p(x)||q(x)) + \max_{x}D_{TV}(p(y|x)||q(y|x))$$

$$\tag{15}$$

$\square$

**Lemma 3** (Integral Probability Metrics). *Consider a measurable space$(\mathcal{X}, \Sigma)$. The integral probability metric associated with a class $\mathcal{F}$ of real-valued functions on $\mathcal{X}$ is defined as*

$$d_{\mathcal{F}}(P,Q) = \sup_{f\in\mathcal{F}}|\mathbb{E}_{X\sim P}[f(X)] - \mathbb{E}_{Y\sim Q}[f(Y)]| \tag{16}$$

*where P and Q are probability measures on $\mathcal{X}$. We demonstrate the following special cases:*

*(a) If $\mathcal{F} = \{f : ||f||_{\infty} \leq c\}$, then we have*

$$d_{\mathcal{F}}(P,Q) = cD_{TV}(P||Q) \tag{17}$$

*(b) If $\mathcal{F}$ is the set of $L-$ Lipschitz function with a norm $||\cdot||$, then we have*

$$d_{\mathcal{F}}(P,Q) = LW_1(P,Q) \tag{18}$$

*In our paper, to distinguish the dynamic transition function, we choose $\mathcal{F}$ to be the class covering $V_M^{\pi}$. Since the value function can converge to $\frac{r_{max}}{1-\gamma}$, it only needs to satisfy the $L_v$-Lipschitz continuity and thus we can get $\frac{r_{max}}{1-\gamma}D_{TV}(p_M||p_{M'}) = L_v W_1(p_M, p_{M'})$ for any arbitrary model $M, M'$.*

## C   Missing Proof

**Theorem 1** (Performance Difference Bound Decomposition). *Let $M_i \in \mathcal{M}$ be the evaluated model and $\pi_i \in \Pi$ be the policy derived from the model. The performance difference bound can be decomposed into three terms,*

$$V^{\pi_2|M_2} - V^{\pi_1|M_1} = (V^{\pi_2|M_2} - V_{M_2}^{\pi_2}) - (V^{\pi_1|M_1} - V_{M_1}^{\pi_1}) + (V_{M_2}^{\pi_2} - V_{M_1}^{\pi_1}) \tag{19}$$

*Proof:* We introduce two additional terms $V_{M_1}^{\pi_1}$ and $V_{M_2}^{\pi_2}$ that allow the performance difference bound objective to be divided into three operators based on the return bounds, which can be reformulated separately.

$$V^{\pi_2|M_2} - V^{\pi_1|M_1} = V^{\pi_2|M_2} - V^{\pi_1|M_1} + (V_{M_1}^{\pi_1} - V_{M_1}^{\pi_1}) + (V_{M_2}^{\pi_2} - V_{M_2}^{\pi_2})$$

$$= (V^{\pi_2|M_2} - V_{M_2}^{\pi_2}) - (V^{\pi_1|M_1} - V_{M_1}^{\pi_1}) + (V_{M_2}^{\pi_2} - V_{M_1}^{\pi_1}) \tag{20}$$

$\square$

**Theorem 2** (Return Bound). *Let $R_{max}$ denote the bound of the reward function, $\epsilon_{\pi}$ denote $\max_s D_{TV}(\pi_1||\pi_2)$ and $\epsilon_{M_1}^{M_2}$ denote $\mathbb{E}_{(s,a)\sim d_{M_1}^{\pi_1}}[D_{TV}(p_{M_1}||p_{M_2})]$. For two arbitrary policies $\pi_1, \pi_2 \in \Pi$, the expected return under two arbitrary models $M_1, M_2 \in \mathcal{M}$ can be bounded as,*

$$V_{M_2}^{\pi_2} - V_{M_1}^{\pi_1} \geq -2R_{max}(\frac{\epsilon_{\pi}}{(1-\gamma)^2} + \frac{\gamma}{(1-\gamma)^2}\epsilon_{M_1}^{M_2}) \tag{21}$$

*Proof:* We give the thorough proof referring to Lemma B.4 in MBPO [20] as follows.

$$V_{M_2}^{\pi_2} - V_{M_1}^{\pi_1} = \sum_{t=0}^{\infty} \gamma^t \sum_{s,a} (p_{t,M_2}^{\pi_2}(s,a) - p_{t,M_1}^{\pi_1}(s,a))r(s,a)$$

$$\geq -R_{max} \sum_{t=0}^{\infty} \gamma^t \sum_{s,a} |p_{t,M_2}^{\pi_2}(s,a) - p_{t,M_1}^{\pi_1}(s,a)| \tag{22}$$

$$= -2R_{max} \sum_{t=0}^{\infty} \gamma^t D_{TV}(p_{t,M_1}^{\pi_1}(s,a)||p_{t,M_2}^{\pi_2}(s,a))$$

According to the Lemma 2, we have:

$$D_{TV}(p_{t,M_1}^{\pi_1}(s,a)||p_{t,M_2}^{\pi_2}(s,a)) \leq D_{TV}(p_{t,M_1}^{\pi_1}(s)||p_{t,M_2}^{\pi_2}(s)) + \max_s D_{TV}(\pi_1(\cdot|s)||\pi_2(\cdot|s))$$
$$= D_{TV}(p_{t,M_1}^{\pi_1}(s)||p_{t,M_2}^{\pi_2}(s)) + \epsilon_\pi \tag{23}$$

Further we expand the first term:

$$D_{TV}(p_{t,M_1}^{\pi_1}(s)||p_{t,M_2}^{\pi_2}(s))$$

$$= \frac{1}{2} \sum_s |p_{t,M_1}^{\pi_1}(s) - p_{t,M_2}^{\pi_2}(s)|$$

$$= \frac{1}{2} \sum_s |\sum_{s'} p_{M_1}^{\pi_1}(s|s')p_{t-1,M_1}^{\pi_1}(s') - p_{M_2}^{\pi_2}(s|s')p_{t-1,M_2}^{\pi_2}(s')|$$

$$\leq \frac{1}{2} \sum_s \sum_{s'} |p_{M_1}^{\pi_1}(s|s')p_{t-1,M_1}^{\pi_1}(s') - p_{M_2}^{\pi_2}(s|s')p_{t-1,M_2}^{\pi_2}(s')|$$

$$\leq \frac{1}{2} \sum_{s,s'} p_{t-1,M_1}^{\pi_1}(s')|p_{M_1}^{\pi_1}(s|s') - p_{M_2}^{\pi_2}(s|s')| + p_{M_2}^{\pi_2}(s|s')|p_{t-1,M_1}^{\pi_1}(s') - p_{t-1,M_2}^{\pi_2}(s')|$$

$$= \frac{1}{2} \mathbb{E}_{s' \sim p_{t-1,M_1}^{\pi_1}(s')}[\sum_s |p_{M_1}^{\pi_1}(s|s') - p_{M_2}^{\pi_2}(s|s')|] + D_{TV}(p_{t-1,M_1}^{\pi_1}(s')||p_{t-1,M_2}^{\pi_2}(s'))$$

$$= \frac{1}{2} \sum_{t'=1}^{t} \mathbb{E}_{s' \sim p_{t'-1,M_1}^{\pi_1}(s')}[\sum_s |p_{M_1}^{\pi_1}(s|s') - p_{M_2}^{\pi_2}(s|s')|] \tag{24}$$

$$= \frac{1}{2} \sum_{t'=1}^{t} \mathbb{E}_{s' \sim p_{t'-1,M_1}^{\pi_1}(s')}[\sum_s |\sum_a p_{M_1}^{\pi_1}(s,a|s') - p_{M_2}^{\pi_2}(s,a|s')|]$$

$$\leq \frac{1}{2} \sum_{t'=1}^{t} \mathbb{E}_{s' \sim p_{t'-1,M_1}^{\pi_1}(s')}[\sum_{s,a} |p_{M_1}^{\pi_1}(s,a|s') - p_{M_2}^{\pi_2}(s,a|s')|]$$

$$= \sum_{t'=1}^{t} \mathbb{E}_{s' \sim p_{t'-1,M_1}^{\pi_1}(s')} D_{TV}(p_{M_1}^{\pi_1}(s,a|s')||p_{M_2}^{\pi_2}(s,a|s'))$$

$$\leq \sum_{t'=1}^{t} \mathbb{E}_{s' \sim p_{t'-1,M_1}^{\pi_1}(s')}[\epsilon_\pi + \mathbb{E}_{a \sim \pi_1}[D_{TV}(p_{M_1}(s|s',a)||p_{M_2}(s|s',a))]]$$

$$= t\epsilon_\pi + \sum_{t'=1}^{t} \mathbb{E}_{s',a \sim p_{t'-1,M_1}^{\pi_1}(s',a)} D_{TV}(p_{M_1}(s|s',a)||p_{M_2}(s|s',a))$$

Then move the result of Eq.(24) to Eq.(23), we can get:

$$D_{TV}(p_{t,M_1}^{\pi_1}(s,a)||p_{t,M_2}^{\pi_2}(s,a)) \leq (t+1)\epsilon_\pi + \sum_{t'=1}^{t} \mathbb{E}_{s',a \sim p_{t'-1,M_1}^{\pi_1}(s',a)} D_{TV}(p_{M_1}(s|s',a)||p_{M_2}(s|s',a))$$
$$\tag{25}$$

Next, we move the result of Eq.(25) to Eq.(22), we can get:

$$V_{M_2}^{\pi_2} - V_{M_1}^{\pi_1}$$

$$\geq -2R_{max}\sum_{t=0}^{\infty}\gamma^t((t+1)\epsilon_\pi + \sum_{t'=1}^{t}\mathbb{E}_{s',a\sim p_{t'-1,M_1}^{\pi_1}(s',a)}D_{TV}(p_{M_1}(s|s',a)||p_{M_2}(s|s',a)))$$

$$= -2R_{max}(\frac{\epsilon_\pi}{(1-\gamma)^2} + \frac{1}{1-\gamma}\sum_{t=1}^{\infty}\gamma^t\mathbb{E}_{s',a\sim p_{t-1,M_1}^{\pi_1}(s',a)}D_{TV}(p_{M_1}(s|s',a)||p_{M_2}(s|s',a))) \tag{26}$$

Here, we first simplify the second term of the Eq.(26)

$$\frac{1}{1-\gamma}\sum_{t=1}^{\infty}\gamma^t\mathbb{E}_{s',a\sim p_{t-1,M_1}^{\pi_1}(s',a)}D_{TV}(p_{M_1}(s|s',a)||p_{M_2}(s|s',a))$$

$$= \frac{\gamma}{1-\gamma}\sum_{t=0}^{\infty}\gamma^t\mathbb{E}_{s',a\sim p_{t,M_1}^{\pi_1}(s',a)}D_{TV}(p_{M_1}(s|s',a)||p_{M_2}(s|s',a))$$

$$= \frac{\gamma}{(1-\gamma)^2}(1-\gamma)\sum_{t=0}^{\infty}\gamma^t\mathbb{E}_{s',a\sim p_{t,M_1}^{\pi_1}(s',a)}D_{TV}(p_{M_1}(s|s',a)||p_{M_2}(s|s',a)) \tag{27}$$

$$= \frac{\gamma}{(1-\gamma)^2}\mathbb{E}_{s',a\sim d_{M_1}^{\pi_1}(s',a)}D_{TV}(p_{M_1}(s|s',a)||p_{M_2}(s|s',a))$$

$$= \frac{\gamma}{(1-\gamma)^2}\epsilon_{M_1}^{M_2}$$

Then we bring this result back to Eq.(26) and the proof is complete.

$$V_{M_2}^{\pi_2} - V_{M_1}^{\pi_1} \geq -2R_{max}(\frac{\epsilon_\pi}{(1-\gamma)^2} + \frac{\gamma}{(1-\gamma)^2}\epsilon_{M_1}^{M_2}) \tag{28}$$

$\square$

**Theorem 3** (Decomposition TVD Bound). *Let $\epsilon_{M_i}^{\pi_i}$ denote $\mathbb{E}_{(s,a)\sim d_{M_i}^{\pi_i}}[D_{TV}(p_{M_i}||p_{M_*})]$. Let $M_i \in \mathcal{M}$ be the evaluated model and $\pi_i \in \Pi$ be the policy derived from the model. The decomposition terms can be bounded as,*

$$V^{\pi_2|M_2} - V^{\pi_1|M_1} \geq \frac{2R_{max}\gamma}{(1-\gamma)^2}(\epsilon_{M_1}^{\pi_1} - \epsilon_{M_2}^{\pi_2} - \epsilon_{M_1}^{M_2}) - \frac{2R_{max}\epsilon_\pi}{(1-\gamma)^2} \tag{29}$$

*Proof:* According to CMLO [21] and Eq.(21), the term $V^{\pi_1|M_1} - V_{M_1}^{\pi_1}$ can be approximated as $-\frac{2R_{max}\gamma}{(1-\gamma)^2}\epsilon_{M_1}^{\pi_1}$, thus we only need to bound the remaining two terms.

For the term $V^{\pi_2|M_2} - V_{M_2}^{\pi_2}$, we use Eq.(21) to bound it.

$$V^{\pi_2|M_2} - V_{M_2}^{\pi_2} \geq -2R_{max}(\frac{\max\limits_{s} D_{TV}(\pi_2||\pi_2)}{(1-\gamma)^2} + \frac{\gamma}{(1-\gamma)^2}\epsilon_{M_2}^{\pi_2})$$

$$= -\frac{2R_{max}\gamma}{(1-\gamma)^2}\epsilon_{M_2}^{\pi_2} \tag{30}$$

Similarly, for the term $V_{M_2}^{\pi_2} - V_{M_1}^{\pi_1}$, we can get:

$$V_{M_2}^{\pi_2} - V_{M_1}^{\pi_1} \geq -2R_{max}(\frac{\epsilon_\pi}{(1-\gamma)^2} + \frac{\gamma}{(1-\gamma)^2}\epsilon_{M_1}^{M_2}) \tag{31}$$

We now combine these three bounds together and complete the proof.

$$V^{\pi_2|M_2} - V^{\pi_1|M_1} = (V^{\pi_2|M_2} - V_{M_2}^{\pi_2}) - (V^{\pi_1|M_1} - V_{M_1}^{\pi_1}) + (V_{M_2}^{\pi_2} - V_{M_1}^{\pi_1})$$

$$\geq -\frac{2R_{max}\gamma}{(1-\gamma)^2}\epsilon_{M_2}^{\pi_2} + \frac{2R_{max}\gamma}{(1-\gamma)^2}\epsilon_{M_1}^{\pi_1} - 2R_{max}(\frac{\epsilon_\pi}{(1-\gamma)^2} + \frac{\gamma}{(1-\gamma)^2}\epsilon_{M_1}^{M_2}) \tag{32}$$

$$= \frac{2R_{max}\gamma}{(1-\gamma)^2}(\epsilon_{M_1}^{\pi_1} - \epsilon_{M_2}^{\pi_2} - \epsilon_{M_1}^{M_2}) - \frac{2R_{max}\epsilon_\pi}{(1-\gamma)^2}$$

$\square$

**Theorem 4** (Unified Model Shift and Model Bias Bound). *Let $\kappa$ denote the constant $\frac{2R_{max}}{(1-\gamma)^2}$ and $\Delta$ denotes $\mathbb{E}_{(s,a)\sim d_{M_1}^{\pi_1}}[D_{TV}(p_{M_2}||p_{M^*})] - \mathbb{E}_{(s,a)\sim d_{M_2}^{\pi_2}}[D_{TV}(p_{M_2}||p_{M^*})]$. Let $M_i \in \mathcal{M}$ be the evaluated model and $\pi_i \in \Pi$ be the policy derived from the model. The unified model shift and model bias bound can be derived as,*

$$V^{\pi_2|M_2} - V^{\pi_1|M_1}$$
$$\geq \kappa(\gamma(\mathbb{E}_{(s,a)\sim d_{M_1}^{\pi_1}}[D_{TV}(p_{M_1}||p_{M^*}) - D_{TV}(p_{M_1}||p_{M_2}) - D_{TV}(p_{M_2}||p_{M_*})] + \Delta) - \epsilon_\pi) \tag{33}$$

*Proof:* Based on Eq.(29), we add a new term $\kappa\mathbb{E}_{(s,a)\sim d_{M_1}^{\pi_1}}[D_{TV}(p_{M_2}||p_{M^*})]$ to reformulate the optimization objective. $\qquad\square$

**Theorem 5** ($|\Delta|$ Upper Bound). *Let $M_i \in \mathcal{M}$ be the evaluated model and $\pi_i \in \Pi$ be the policy derived from the model. The term $\Delta$ can be upper bounded as:*

$$|\Delta| \leq \frac{2\gamma}{1-\gamma}\mathbb{E}_{(s,a)\sim d_{M_1}^{\pi_1}}[D_{TV}(p_{M_1}||p_{M_2})\max_{s,a} D_{TV}(p_{M_2}||p_{M^*})] + \frac{2\epsilon_\pi}{1-\gamma}\max_{s,a} D_{TV}(p_{M_2}||p_{M^*}) \tag{34}$$

*Proof:* First, we combine these two terms.

$$|\Delta| = |\mathbb{E}_{(s,a)\sim d_{M_1}^{\pi_1}}[D_{TV}(p_{M_2}||p_{M^*})] - \mathbb{E}_{(s,a)\sim d_{M_2}^{\pi_2}}[D_{TV}(p_{M_2}||p_{M^*})]|$$

$$= (1-\gamma)|\sum_{t=0}^{\infty}\gamma^t\sum_{s,a}(p_{t,M_1}^{\pi_1}(s,a) - p_{t,M_2}^{\pi_2}(s,a))D_{TV}(p_{M_2}(s'|s,a)||p_{M^*}(s'|s,a))|$$

$$\leq (1-\gamma)\max_{s,a} D_{TV}(p_{M_2}(s'|s,a)||p_{M^*}(s'|s,a))\sum_{t=0}^{\infty}\gamma^t\sum_{s,a}|p_{t,M_1}^{\pi_1}(s,a) - p_{t,M_2}^{\pi_2}(s,a)| \tag{35}$$

$$= 2(1-\gamma)\max_{s,a} D_{TV}(p_{M_2}||p_{M^*})\sum_{t=0}^{\infty}\gamma^t D_{TV}(p_{t,M_1}^{\pi_1}(s,a)||p_{t,M_2}^{\pi_2}(s,a))$$

Recalling that we get the result of the sum equation above in Eq.(25), and then we have:

$$|\Delta| \leq 2(1-\gamma)\max_{s,a} D_{TV}(p_{M_2}||p_{M^*})(\frac{\epsilon_\pi}{(1-\gamma)^2} + \frac{\gamma}{(1-\gamma)^2}\epsilon_{M_1}^{M_2})$$

$$= \frac{2\gamma}{1-\gamma}\mathbb{E}_{s,a\sim d_{M_1}^{\pi_1}}[D_{TV}(p_{M_1}||p_{M_2})\max_{s,a} D_{TV}(p_{M_2}||p_{M^*})] + \frac{2\epsilon_\pi}{1-\gamma}\max_{s,a} D_{TV}(p_{M_2}||p_{M^*}) \tag{36}$$

$$\square$$

# D  Experimental Details

## D.1  Environment Setup

We evaluate the algorithm over a series of MuJoCo [46] continuous control benchmark tasks. To ensure fairness, we use the standard 1000-step version of all the environments. The details of the environment setup are from OpenAI Gym [3], as shown in Table 1.

Table 1: The general outline of the MuJoCo environment.

| Environment-Version | State Dim | Action Dim | Termination |
|---|---|---|---|
| Ant-v2 | 27 | 8 | obs[0]<0.2 or obs[0] > 1.0 |
| HalfCheetah-v2 | 17 | 6 | - |
| Hopper-v2 | 11 | 3 | obs[1] $\geq$ 0.2 or obs[0] $\leq$ 0.7 |
| Humanoid-v2 | 45 | 17 | obs[0] < 1.0 or obs[0] > 2.0 |
| InvertedPendulum-v2 | 4 | 1 | obs[1] > 0.2 or obs[1] < -0.2 |
| Walker2d-v2 | 17 | 6 | obs[0] $\geq$ 2.0 or obs[0] $\leq$ 0.8 or obs[1] $\geq$ 1.0 or obs[1] $\leq$ -1.0 |

## D.2 Baseline implementation

**MFRL Baselines.** We use two state-of-the-art model-free algorithms, i.e. SAC [15] and PPO [40], to do baseline comparison. To demonstrate the final performance and sampling efficiency of our method, we train SAC for 3M steps, which is much more than MBRL algorithms. The hyperparameters are consistent with the author's settings.

**MBRL Baselines.** We use several state-of-the-art model-based algorithms to do baseline comparison, covering CMLO [21], MBPO [20], SLBO [31] and STEVE [4]. The implementation of CMLO is based on the opensource repo published by the author and all of the hyperparameters are set according to the paper [21]. Our algorithm USB-PO is implemented based on the opensource repo published by Janner who is the author of MBPO.

We present the final performance on six continuous benchmark tasks in Table 2. The results demonstrate that our algorithm achieves competitive performance compared to both MBRL and MFRL baselines over these tasks. Each result in the table shows the average and standard deviation on the maximum average returns among different random seeds and we choose 250K for HalfCheetah-v2, 300K for Walker2d-v2, 300K for Humanoid-v2, 250K for Ant-v2, 15K for Inverted-Pendulum-v2, 120K for Hopper-v2.

Table 2: The final performance on six continuous benchmark tasks.

|  |  | HalfCheetah | Humanoid | Walker2d |
|---|---|---|---|---|
|  | STEVE | 12406.29±458.08 | 4318.32±853.60 | 1109.23±1163.74 |
|  | SLBO | 1915.47±1398.73 | 459.46±34.27 | 3107.93±1887.09 |
| MBRL | MBPO | 12765.67±594.54 | 5546.77±221.72 | 4582.06±67.44 |
|  | CMLO | 10143.55±193.82 | 5577.01±219.89 | 4807.60±99.89 |
|  | USB-PO | **15105.91±177.75** | **5973.75±110.99** | **5691.62±162.57** |
| MFRL(@3M steps) | SAC | 15012 | 6207 | 5879 |
|  |  | Ant | InvertedPendulum | Hopper |
|  | STEVE | 779.72±45.67 | 778.54±265.51 | 1131.61±623.52 |
|  | SLBO | 707.79±218.80 | 793.24±334.90 | 898.68±233.21 |
| MBRL | MBPO | 4926.10±818.38 | 1000.00±0.00 | 3436.00±120.72 |
|  | CMLO | 5123.71±783.97 | 1000.00±0.00 | 3495.41±71.02 |
|  | USB-PO | **6340.84±119.06** | **1000.00±0.00** | **3694.22±46.19** |
| MFRL(@3M steps) | SAC | 5934 | 1000 | 3610 |

## D.3 Hyperparameters

Our algorithm USB-PO is based on MBPO [20] and is implemented according to the opensource repo published by the MBPO author. Except for the learning rate in phase 2 of our USB-PO algorithm, the hyperparameters are completely identical to the MBPO settings for all environments. In all benchmark tasks, we set this learning rate to 1e-4.

## D.4 Computing Infrastructure

In Table 3, we list our computing infrastructure and the computational time for training USB-PO on these six continuous benchmark tasks. Note that the time we report is the cost for 4 random seeds simultaneously on one graphics card. For Humanoid, only two random seeds can be run simultaneously because of the limitation of graphics memory.

Table 3: Computing infrastructure and the computational time for each benchmark task compared to MBPO, where the time unit d denotes day and h denotes hour.

|  | HalfCheetah | Humanoid | Walker2d | Ant | InvertedPendulum | Hopper |
|---|---|---|---|---|---|---|
| CPU | AMD EPYC 7B12 64-Core Processor | | | | | |
| GPU | NVIDIA 2080Ti | | | | | |
| MBPO times | 2.46d | 1.64d | 1.75d | 2.88d | 3.43h | 17.81h |
| USB-PO times | 2.29d | 1.51d | 1.65d | 2.91d | 3.42h | 18.28h |

# E    Comparison with Prior Works

In this section, we compare USB-PO with prior theoretical works to emphasize our contribution, as a complementary to the main paper. First, we give a summary and then show the details as follows. MBPO-Style does not consider model shift and CMLO-Style rely on a fixed threshold to constrain model shift. Our algorithm, USB-PO, adaptively adjusts the model updates in a unified manner (unify model shift and model bias) to get the performance improvement guarantee.

**MBPO-Style [20, 41, 24, 53].**    They use the return discrepancy bound $V^{\pi|M} \geq V_M^\pi - C(\epsilon_m, \epsilon_\pi)$ to improve the lower bound on the performance under the real environment, i.e. as long as improving $V_M^\pi$ by more than $C(\epsilon_m, \epsilon_\pi)$ can guarantee improvement on $V^{\pi|M}$. Obviously, This scheme is guaranteed under a fixed model and it does not consider the change in model dynamic during updates nor the performance variation concerning model shift. Even worse, if the model has some excessive updates, it is impractical to find a feasible solution to meet the improvement guarantee.

**CMLO-Style [21].**    They use the performance difference bound under the model-based setting $V^{\pi_2|M_2} - V^{\pi_1|M_1} \geq C$ to directly consider model shift and model bias. However, they finally derive a constrained lower-bound optimization problem and use a fixed threshold to constrain model shift, i.e. $\sup_{s \in \mathcal{S}, a \in \mathcal{A}} D_{TV}(P_{M_1}(\cdot|s,a) \| P_{M_2}(\cdot|s,a)) \leq \sigma_{M_1, M_2}$ and determine when to update the model accordingly. Notably, we find that this fixed threshold plays a key role in the whole algorithm and needs to be carefully adjusted for each environment. If this threshold is set too low, the model bias of the following iteration will be large, which impairs the subsequent optimization process. If this threshold is set too high, the performance improvement can no longer be guaranteed. Additionally, using a fixed threshold during the whole training process makes the algorithm problematic to adjust adaptively.

**USB-PO (Ours).**    Following CMLO-Style [21], we also use the performance difference bound under the model-based setting to directly consider model shift and model bias. Compared to relying on a fixed threshold to constrain model shift, we use a transformation to unify model shift and model bias into one formulation without the constraint (Theorem 4). Due to the intractable property of $\Delta$, we further explore the upper bound of $|\Delta|$, finding that $\Delta$ can be ignored with respect to model shift and model bias alone (Theorem 5). Finally, the optimization objective we get can be used to fine-tune $M_2$ in a unified manner to adaptively adjust the model updates to get a performance improvement guarantee. Notably, our algorithm can use the same learning rate of Phase 2 and our algorithm is robust to this learning rate. To the best of our knowledge, this is the first method that unifies model shift and model bias and adaptively fine-tunes the model updates during the training process.

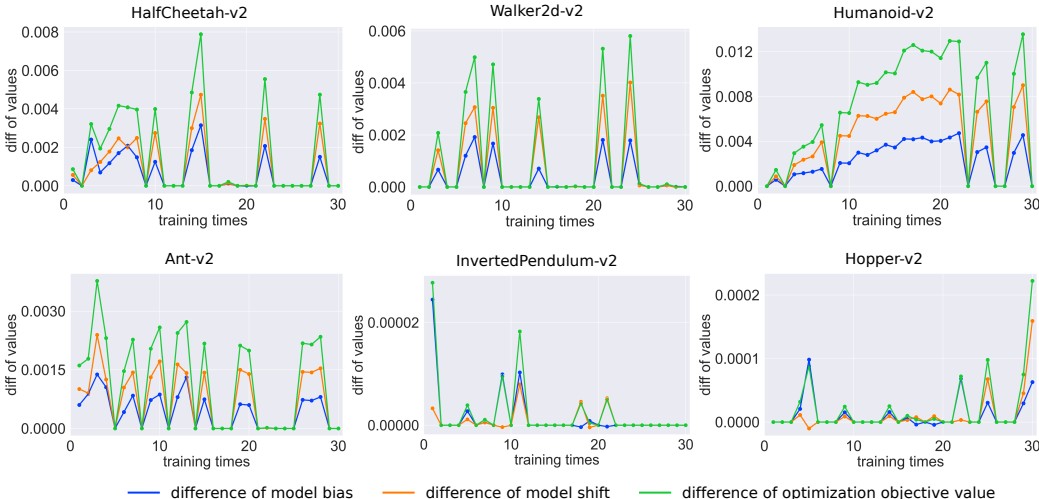

Figure 10: We choose a specific random seed to show the details of the first 30 training times on all benchmark tasks, covering the difference of optimization objective value, the model shift term and the model bias term before and after the fine-tuning process.

# F Additional Experiment

## F.1 Working Mechanism Extension

To illustrate that the ability of USB-PO to reduce both model shift and model bias potentially is not a coincidence that exists only in the Walker2d environment, we add experimental results in other MuJoCo [46] environments. As shown in Figure 10, when the fine-tuning actually operates, the difference of the model shift term and the model bias term among all of the benchmark tasks are generally both positive, further validating our superiority.

## F.2 Ablation Study Extension

Here, we show the results of the ablation study on all of the MuJoCo benchmark tasks.

As shown in Figure 11, only optimizing the model shift term results in a drop in sample efficiency while only optimizing the model bias term leads to performance deterioration. Only fine-tuning the model updates in a unified manner can achieve excellent performance.

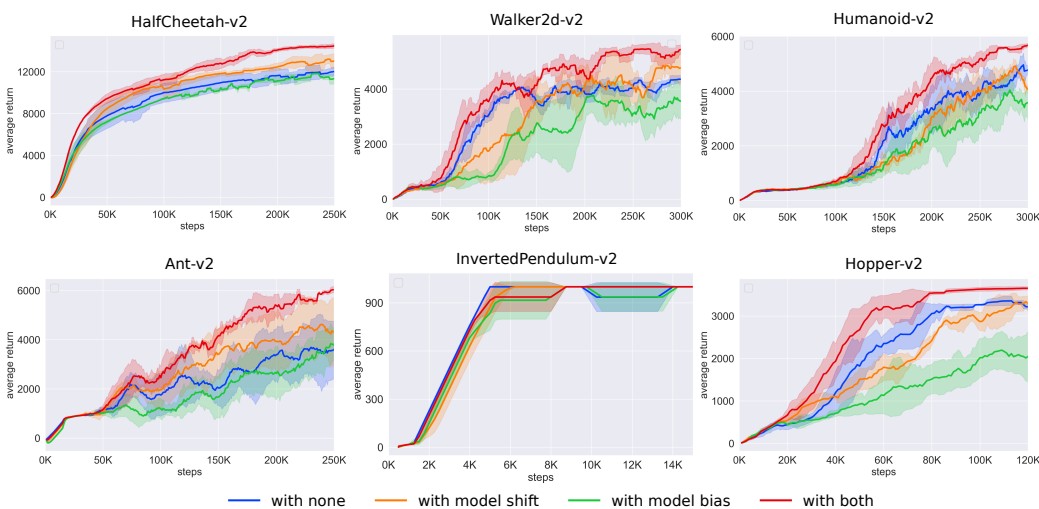

Figure 11: **Optimization Objective Variants** on all MuJoCo benchmark tasks.

## F.3 Ensemble numbers

To illustrate the justification of the ensemble numbers we set, we conduct the ablation experiment on more ensemble numbers containing 3, 5, and 7. As Figure 12 shows, as the number of ensemble models goes up, the performance will be higher and more stable, but it will cost more time. To maintain the balance between performance and time, we finally set the value of this parameter to 7, which is recommended by the MBPO [20] original repo.

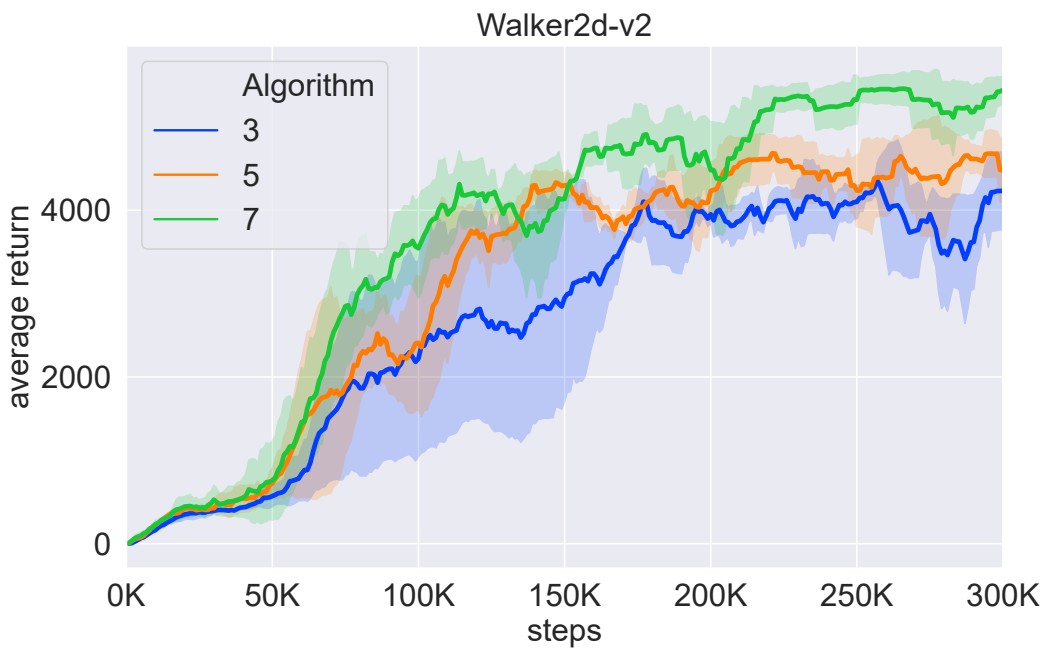

Figure 12: Ablation experimental results of the ensemble model numbers.

