# OpenReview forum: "How to Fine-tune the Model: Unified Model Shift and Model Bias Policy Optimization"
_NeurIPS.cc/2023/Conference — NeurIPS 2023 poster_

### Official Review · Reviewer_eAwQ · 2023-06-30

**Soundness:** 3 good
**Presentation:** 3 good
**Contribution:** 3 good
**Rating:** 5
**Confidence:** 4

**Summary:**

This paper investigates the problem of model shift in model-based Reinforcement Learning (RL). The author proposes a two-stage model learning method called USB-PO (Unified model Shift and model Bias Policy Optimization). The first stage is the same as previous work, updating based on the previous model using model replay buffer. The second stage is based on the previous model and the replay buffer, aiming to simultaneously reduce model shift and model bias after the model first stage update. The paper also provides a theoretical analysis for the proposed method, and the effectiveness of the method is demonstrated in experiments conducted on MuJoCo.

**Strengths:**

1. As far as I know, the method proposed in this paper is novel.
2. The authors provide theoretical analysis and motivation for their method.
3. The ablation study in the paper is rather comprehensive.

**Weaknesses:**

1. What is the basis for lines 133-135? I don't think such a conclusion can be derived from Definition 1 and Theorem 1.
2. What is the relationship between Theorem 2 and 3? Both seem to be proving different bounds for the same term, what is the significance of this? Judging from the appendix, there appears to be a typo in Theorem 3.
3. In the experiment section, the performance curve of the MBRL method CMLO drops much more than what is shown in the original paper. I would like to know the reason for this.
4. The baselines STEVE and SLBO are too outdated, and they can no longer be considered as state-of-the-art (SOTA) baselines for MBRL.

**Questions:**

1. The two-stage model learning will inevitably lead to more time consumption. Can you specify how much the time consumption has increased?
2. Please compare with some recent MBRL methods instead of STEVE and SLBO, such as PDML [1] and ALM [2].


Reference:

[1]. Wang et. al. "Live in the Moment: Learning Dynamics Model Adapted to Evolving Policy," in ICML 2023, https://arxiv.org/abs/2207.12141.

[2]. Ghugare et. al. "Simplifying Model-based RL: Learning Representations, Latent-space Models, and Policies with One Objective," in ICLR 2023, https://arxiv.org/abs/2209.08466.

**Limitations:**

1. The time efficiency needs to be reported.
2. More baseline comparisons are required.
3. It would be desirable to compare with more benchmarks, such as DMC and Metaworld, not just MuJoCo.

---

> ### Author Rebuttal · Authors · 2023-08-09
>
> We would like to appreciate the reviewer for the recognition and valuable comments! Our specific responses to the questions raised by the reviewer are as follows:
>
> ___
>
> ## 1. The basis for lines 133-135
> According to the previous work MBPO, the return discrepancy scheme refers to the difference between the return under the model and that in the real environment. Under our notation, the return discrepancy is denoted as $V^{\pi_i|M_i}-V^{\pi_i}\_{M_i}$. According to Theorem 1, we make some revisions on line 133-135 to make it clear. **Please refer to point 2 in global response.**
>
> ___
>
> ## 2. The relationship between Theorem 2 and 3
> Sorry, we do have a typo in Theorem 3. We correct the formula in Theorem 3 (Eq.(7)). **Please refer to point 3 in global response.**
>
> ___
>
> ## 3. The performance curve of the MBRL method CMLO drops much more than what is shown in the original paper.
> We at first ran a lot of random seeds for CMLO when verifying its performance, and found that in very occasional cases CMLO would have extremely high performance (e.g. ant for 8088, walker2d for 7553). We do not doubt the performance of CMLO, we just think that the authors just select such special cases in their plots under limited runs. **For a fair comparison, we exclude these special cases from our plots.** Moreover, the HalfCheetah and Ant environments are only plotted up to 250K in our experiment, which may seem to have lower performance, but is actually compatible with CMLO, **as we take greater smoothing operation**.
>
> ___
>
> ## 4. More baseline comparisons
> We are sorry for being unable to run PDML because the repo released by the author misses the key source-code file, **so we contact the author to get the results in their original paper**. **Please refer to Figure 2 in global response pdf.** USB-PO still exhibits SOTA asymptotic performance and sample efficiency, although not as good as PDML in Humanoid environments. We are not claiming to beat all SOTA algorithms with sophisticated designs as PDML needs extra computational resources to compute policy shift ($ξ_{\pi_i}$) and to memorize the policy sequence. Although the growth on time reported in the PDML is small compared to the MBPO, **our method USB-PO has the potential to shorten the time consumption, please refer to the time cost in Appendix C.4**. As for the ALM, the training for its classifier may introduce instability, which impairs the performance.
>
> ___
>
> ## 5. Can you specify how much the time consumption has increased? / Time efficiency needs to be reported.
> According to Section 5.2, although USB-PO is a two-stage model training process, **continuing to use the fine-tuned model for the next iteration has the potential to accelerate model convergence** (the fine-tuning process reduces model bias) and then possibly reduce the training time. **Please refer to the time cost we reported compared to MBPO in Appendix C.4**. Training time is reduced on Humanoid, HalfCheetah, and Walker2d.
>
> In addition, we make some revisions to describe the computational cost. **Please refer to point 7 in global response.**
>
> ___
>
> ## 6. It would be desirable to compare with more benchmarks, such as DMC and Metaworld, not just MuJoCo.
> Sorry, we don't give experimental results on DMC or Metaworld within a limited time for the following three reasons:
>
> · Previous methods including MBPO, M2AC, P2P, BMPO, CMLO, and so on have not been experimented on DMC or Metaworld, so DMC and Metaworld are unfamiliar to us.
>
> · After investigating DMC and Metaworld, we made an attempt and found that DMC or Metaworld is quite different from MuJoCo, so both baseline and USB-PO need to tune the parameters carefully.
>
> · MBRL algorithms generally consume a long time to train once, e.g. almost 3 days for MBPO and 6 days for P2P on HalfCheetah.
>
> Though we don’t give specific results, we set this as future work.
>
> ___
>
> We expect the reviewer could increase the score if we address the issues the reviewer raised. Of course, if there are more questions, we are willing to further discuss them with the reviewer.

---

> > ### Comment · Reviewer_eAwQ · 2023-08-20
> >
> > Thanks for the author's response. I still believe that the reasons mentioned by the author are not excuses for not conducting experiments in other environments, so I will maintain my score

---

> > > ### Author Response · Authors · 2023-08-20
> > > **Discussion reply to the reviewer eAwQ**
> > >
> > > Dear reviewer,
> > >
> > > We sincerely thank you for your reply! We will take experiments on other environments (DMC/Metaworld) as future work. However, we would like to emphasize that the aim of this paper is to lay down the theoretical foundations of the extra design of phase 2. **Our experiments are designed to validate the superiority of unifying model shift and model bias and show its working mechanism. The results shown in the experiment are sufficient to support our theory.** In addition, during the rebuttal period, we add ALM and PDML and analyze the time cost to further strengthen our method. While we intend to evaluate the proposed algorithm on an extensive deep RL benchmark in the future (DMC/Metaworld), these experiments are beyond the scope of the current work. We sincerely wish you to value our work and reconsider the score.
> > >
> > > Best wishes!
> > >
> > > The authors.

---

### Official Review · Reviewer_VRuX · 2023-07-06

**Soundness:** 3 good
**Presentation:** 1 poor
**Contribution:** 3 good
**Rating:** 3
**Confidence:** 3

**Summary:**

This paper proposes a new MBRL algorithm which adaptively adjusts the model update by considering the model shift (distance towards previous step update) and the model bias (towards true env model) simultaneously, which avoids model overfitting and shows better empirical performance compared to only considering either of them.

**Strengths:**

- This paper clearly discusses the drawbacks of previous MBRL approaches, either considering model bias only, which gives possible excessive model updates, or considering model shift only, which might be subject to overfitting to the previous model. A natural and neat idea is proposed by unifying the two aspects together to further fine-tune the model, which supported by strong empirical results.

- The experiments section is thorough. The ablation study is interesting, which shows the performance of USB-PO is better than removing any of the model bias or shift term. Section 5.2 is also nice, which empirically verifies the assumptions in the method development, by removing the consideration of $\Delta$, as well as the visualization on the optimization objective value to see how the update happens. These experiments facilitate the understanding the method.


**Weaknesses:**

- The biggest weakness of the paper is that it is very poorly written, the idea is simple but the way the paper organizes and writes make it very difficult to convey the main message. For example, Section 4.1 is too general and it is hard to get what it is trying to say. Algorithm 1 does not show any important message, for example, Line 115 says the optimization objective is MLE loss and the phase 2 is for finetuning, which is too high-level and it makes readers confused.

- This might be correlated with point 1, the algorithm in Algorithm is not super clear to me. It seems Equation 3 is used to learning $M_2$, but the Algorithm 2 Line 2 says we get $M_2$ from $M_1$ update, this is confusing. Is Line 3 only for finetuning $M_2$ given the previously learned $M_1$, $M_2$? These are the main points of the paper and it is important to make things clear and precise.



**Questions:**

- For Line 195, could the $M_1$ ensembles be also used in the bias term estimate?
- The experiment in Figure 6, is it possible to try more lr of the MBPO to showcase the advantage?
- Another ablation study is to varying the number of ensembles for the models, and comment on the computational complexity.

---

> ### Author Rebuttal · Authors · 2023-08-09
>
> We thank the reviewer for identifying our technical contributions. The valuable comments help us improve our submission! The reviewer’s primary concern seems to be how the paper is written and organized. Our specific responses to the concerns raised by the reviewer are as follows:
>
> ___
>
> ## 1. Section 4.1 is too general and it is hard to get what it is trying to say. Algorithm 1 does not show any important message. / The algorithm in Algorithm is not super clear.
>
> We notice a strong correlation between the two points mentioned by the reviewer in the Weaknesses, so **we make a unified answer here**. After carefully checking the reviewer’s comments and reading the statements in our paper, we find that Section 4.1 is indeed not very clear in its description and fails to highlight the main points. Also, the notation $p_M$ in Algorithm 1 and the notation $p_{M_1}$ and $p_{M_2}$ in Section 4.1 do have some confusion for the understanding. Therefore, we make revisions to clarify Section 4.1. Furthermore, we integrate Algorithm 2 into Algorithm 1 to unify the notation, removing the confusing notation $p_M$ and removing the original Algorithm 2 from the main paper. **We invite the reviewer to refer to point 4 and point 5 in global response . Moreover, we add a schematic diagram in global response pdf to further describe our method. Please refer to point 8 in global response.**
>
> Here we provide additional clarifications to address the reviewer’s confusion. Algorithm 1 wants to emphasize the difference from the traditional MBRL algorithms, namely the extra fine-tuning process (phase 2). $M_1$ denotes the model backed up before training by MLE while $M_2$ denotes the learned model after training by MLE. **Due to the possible performance drop, we devise phase 2 to further fine-tune $M_2$. As for $M_1$, it is used for computing the model-shift term in Eq.(3).**
>
> ___
>
> ## 2. could the $M_1$ ensembles be also used in the bias term estimate?
> Model bias refers to the error between the learned model and the real environment. According to previous work, the real environment is usually estimated using all models not selected in the ensemble models. Since **the model bias of $p_{M_2}$ is to be estimated in Eq.(3)**, the calculation only involves the $M_2$ ensemble rather than using the $M_1$ ensemble.
>
> However, **it may be possible that using the** $M_1$ **ensemble could lead to more accurate estimates, but this is not our key point and requires further research to validate.**
>
> ___
>
> ## 3. Figure 6: more lr of the MBPO to showcase the advantage
> We conduct the ablation experiment on more learning rates containing 7e-4, 5e-4, 3e-4, 1e-4 and 1e-3 (the original MBPO) to compare with USB-PO, further strengthening the advantage of USB-PO. **Please refer to Figure 4 in global response pdf.**
>
> ___
>
> ## 4. Another ablation study is to varying the number of ensembles for the models, and comment on the computational complexity.
> We conduct the ablation experiment on more ensemble numbers containing 3, 5 and 7. We find that **as the number of ensemble models goes up, the performance will be higher and more stable, but it will cost more time**. To maintain the balance between performance and time, we finally set the value of this parameter to 7, which is recommended by the MBPO original repo. **We invite the reviewer to refer to Figure 5 and Table 1 in global response pdf.**
>
> ___
>
> If we have addressed the reviewer's concerns, we expect the reviewer to consider improving the score. If the reviewer has any additional questions or comments, we would be happy to discuss them further.

---

> > ### Comment · Reviewer_VRuX · 2023-08-19
> >
> > Thanks for the authors' response. I appreciate the authors' additional experiments on the learning rates and the number of ensemble models, it is indeed helpful. However, i am still concerned about the clarity, presentation and writing of the paper. I personally do think the current version is not ready to be publishable in the conference like Neurips. I recommend that the authors invest some effort into revising the paper. This could also potentially enhance the paper's impact.

---

> > > ### Author Response · Authors · 2023-08-19
> > > **Reply to the reviewer VRuX**
> > >
> > > Dear reviewer,
> > >
> > > We sincerely appreciate your valuable feedback and have taken it into careful consideration. We first would like to recall our writing and presentation as follows.
> > >
> > > · In Section 4.1, we provide a comprehensive overview. We highlight that the novel aspect of our methodology, as discussed in this paper, centers around the design of phase 2. This step aims to unify model shift and model bias, leading to a performance improvement guarantee.
> > >
> > > ·  Following this, in Section 4.2, we provide the theoretical proof about why can get a performance improvement guarantee and the derivation for the optimization objective of phase 2 (Eq. (3)).
> > >
> > >
> > > · Lastly, Section 4.3 is dedicated to the practical implementation of the algorithm.
> > >
> > >
> > > We'd like to emphasize that we've diligently acted upon your suggestions for Section 4.1. We have seamlessly integrated Algorithm 2 into Algorithm 1, removed Algorithm 2 from the main paper, and updated the notation for $p_M, p_{M_1}, p_{M_2}$ to prevent misunderstanding, as noted in points 4 and 5 in the global response. This version diverges from our initial submission.
> > >
> > >
> > > Regarding your mention of additional concerns, we kindly ask you to specify them so that we may continue our revision process. Your continued guidance is immensely appreciated.
> > >
> > >
> > > Best wishes!
> > >
> > > The authors.

---

### Official Review · Reviewer_dkWj · 2023-07-06

**Soundness:** 4 excellent
**Presentation:** 4 excellent
**Contribution:** 3 good
**Rating:** 8
**Confidence:** 4

**Summary:**

The paper considers model based RL (MBRL), focusing on the question of fine-tuning the model while learning the policy (the policy optimization).  While prior works treat these two aspects somewhat separately with tuning thresholds, the authors derive a single cost function that incorporates both.  The paper presents a lengthy derivation of the cost function and shows through inequalities and assumptions that it is a well grounded approach, and then numerical experiments validate the approach and compare to MBRL methods as well as model free (MFRL) methods.

**Strengths:**

The paper is very clearly written, with good motivation and placement of the results in the state of the art.  The idea of combining the model refinement and policy learning is perhaps obvious; what is not obvious is how this can be carried out.  The paper does this theoretically, and then carries this through to a useful algorithm.  The derivations are clear and the assumptions needed to bring this to a numerical method are laid out.

Theorem 2 is nice because it bounds the expected return difference between and 2 models and their respective policies.  This should be useful in other learning contexts.

The meta-algorithm (Algorithm 1) puts the overall ideas across in a way that doesn’t require the reader to understand the subtleties of the proofs.


**Weaknesses:**

The reviewer doesn't see any significant weaknesses in the paper.  Some clarifying questions are listed below.

**Questions:**

Perhaps it is worth saying more about the assumption after eqn (9), regarding policy shift being small wrt the model bias?  When would this not be true?

Deriving eqn (10) relies on a Gaussian assumption.  Could you say why this is generally valid, or when it might not be so valid?

Similarly in eqn (11) the Wasserstein distance is easily calculated for the Gaussian case.  So here the Gaussian assumption is carrying through?

After Algorithm 2 statement, “following [18]”: What is the elite set of B models?  And you choose one of these with uniform probability? How large is B typically?

The “pull-back” interpretation is interesting.  That is what you call the no-update case?

In the appendix, Table 2, top 3 cases, the model-free works well, albeit with a lot of steps.  Can you compare the overall complexity of the MFRL and USB-PO for these cases, accounting for cost per step of each?  Also, is there a hybrid MFRL-MBRL approach?



**Limitations:**

There are no negative social issues.

---

> ### Author Rebuttal · Authors · 2023-08-09
>
> We sincerely thank the reviewer for the high recognition of our work! Our specific responses to the questions raised by the reviewer are as follows:
>
> ___
>
> ## 1. Policy shift wrt the model bias
> · According to the experiment in MBPO, when the amount of data provided by the policy for training the model is very small, the policy shift can not be ignored.
>
> · In my view, as the model gradually converges, model bias will drop to a very small amount, which may make the policy shift unignorable. Further research on this topic may improve performance and save computational resources.
> ___
>
> ## 2. The detail of deriving Eq.(10)
> Deriving Eq.(10) does not require the assumption of the Gaussian distribution. Please refer to **line 167-170**, the derivation of Eq.(10) **only needs the assumption that $V^\pi_M$ is** $L_v$**-Lipschitz** and please refer to **Appendix A Lemma 3** for the corresponding details. Though satisfying the Lipschitz continuity is actually difficult, **this occurred frequently in previous theoretical work to get better convergence properties**. Based on the above statement, deriving Eq.(3) also does not require the assumption of Gaussian distribution, except that a closed-form expression can be further obtained with the help of the Gaussian distribution.
>
> **To prevent misunderstanding, we make revisions on line 171-173. Please refer to point 6 in global response.**
> ___
>
> ## 3. Validity of the Gaussian distribution assumption
> In MBRL, it is normal to assume that the model is subject to the Gaussian distribution. Since PETS [6] demonstrated that **Gaussian ensemble models can prevent model overfitting and can better capture model uncertainty** (please refer to the details in **line 180-186**), the subsequent methods (MBPO P2P CMLO M2AC...) have all used this approach. But if you're considering specifically that samples generated by the policy for training the model are internally unbalanced, using the Gaussian distribution has obvious weaknesses.
>
> [6] Kurtland Chua et al. “Deep reinforcement learning in a handful of trials using probabilistic dynamics models”. In: Advances in neural information processing systems 31 (2018).
>
> ___
>
> ## 4. The details of elite set
> The elite set first appeared as a trick in the source code published by the authors of MBPO. Each time the model is trained, a portion of the samples are set aside for holdout validation. Then, the algorithm is ranked from highest to lowest based on the validation error, and the $K$ lowest ranked is set as the elite set ($K<B$). **When generating rollouts, randomly selecting a model from the elite set rather than the original ensemble set gives better performance**. We set the same parameters as MBPO, namely the size of ensemble models is 7 and the size of the elite set is 5.
>
> ___
>
> ## 5. The pullback interpretation
> Thanks again for your confirmation of the “pull-back” explanation! However, this is not for the no-update case. The no-update case means the fine-tuning optimization objective (Eq.(3)) basically does not change the updates of $M_2$ and it can be considered that the updates generated by MLE are appropriate for Eq.(3). When the MLE generates updates with large model shift which may impair the performance, the algorithm can fine-tune (**pull**) the model updates to get the performance improvement (**back**). We call this process of securing the performance improvement guarantee as “pull-back”.
>
> ## 6. Overall complexity
> Sorry, we can not analyze the overall complexity theoretically within the limited time, thus we choose to report the time cost as below and we leave this theoretical work as future work.
>
> | Env | USB-PO | SAC |
> | :-----:|:----: | :----: |
> | HalfCheetah | 2.29day(250K) | 12.53h(3M) |
> | Humanoid | 1.51day(300K) | 12.83h(3M) |
> | Walker2d | 1.65day(300K) | 12.55h(3M) |
>
> As the above table shows, though SAC needs a lot of steps to get asymptotic performance, it costs much less time compared to USB-PO, because USB-PO costs much more time to train the model in each step. **However, in some sample-dangerous scenarios, we have to sacrifice the time complexity (MFRL) to pursue the sample complexity (MBRL)**. As for the hybrid algorithms, as far as we know, when the model is used for generating rollouts, this approach is generally recognized as a model-based algorithm.
> ___
>
> If the reviewer has more questions, please let us know and we'll be happy to answer them!

---

> > ### Comment · Reviewer_dkWj · 2023-08-16
> > **Reply**
> >
> > The reviewer appreciates the authors replies to the questions posed.  Also, the reviewer appreciates the specific changes put forward, including combining algorithm 1 and algorithm 2 statements, and incorporating the additional related work.  I am satisfied with my review rating.

---

> > > ### Author Response · Authors · 2023-08-17
> > > **Thank you for your affirmative reply!**
> > >
> > > Dear reviewer,
> > >
> > > We sincerely thank you for your high recognition of our work!
> > >
> > > Best wishes!
> > >
> > > The authors.

---

### Official Review · Reviewer_C4Te · 2023-07-07

**Soundness:** 3 good
**Presentation:** 3 good
**Contribution:** 3 good
**Rating:** 6
**Confidence:** 3

**Summary:**

This paper studies the learning of the dynamics model in model-based reinforcement learning. The authors propose a novel method USB-PO that has provable properties.

**Strengths:**

1. The paper studies an important problem in model-based RL, namely the model shift and model bias balance during model update without heavy dependence on the threshold and a lack of adaptability.
2. The algorithm is straightforward and the theoretical results also match the expectation.
3. The paper is clearly written and easy to follow.

**Weaknesses:**

1. In the pseudocode Algorithm 1, what is the difference between $M$ and $M_2$? Are they supposed to be the same?
2. Can the authors elaborate the MBRL algorithms that have heavy dependences on the threshold?
3. Can phase 1 and phase 2 be integrated to a single joint optimization objective? It would be easier to implement if the algorithm is a trust-region style algorithm that maintains a single model and regulates the updated and the last-iteration model.
4. There are also previous MBRL works [1, 2, 3] that have a dual update procedure, in a very similar way to how the proposed method regulates the model updates. I recommend the authors to also discuss these related works in a later version of the manuscript.

[1] Zhang et al. Conservative Dual Policy Optimization for Efficient Model-Based Reinforcement Learning.\
[2] Sun et al. Dual Policy Iteration.\
[3] Levine et al. Learning neural network policies with guided policy search under unknown dynamics.

**Questions:**

See the weakness section above.

**Limitations:**

See the weakness section above.

---

> ### Author Rebuttal · Authors · 2023-08-09
>
> We greatly appreciate the reviewer’s insightful comments on our paper. The relevant literature mentioned in your comment helps us better polish the paper! Our specific responses to the questions raised by the reviewer are as follows.
>
> ___
>
> ## 1. The difference between $M$ and $M_2$
>
> $M$ and $M_2$ are different. The purpose of Algorithm 1 is to put the overall ideas that don’t require the reader to understand the following details. Thus, in Algorithm 1 we do not distinguish the model backed up before training with MLE ($M_1$) and the model after training with MLE ($M_2$), but universally use $M$ to denote them. Later, to better illustrate our method theoretically, we denote $M_1$ and $M_2$ separately. Precisely, $M_2$ **denotes the model that needs to be fine-tuned after the training process by MLE in each iteration**.
>
> **To prevent misunderstanding, we revise the Algorithm pseudo-code and Section 4.1. Please refer to point 4 and 5 in global response. Also, we add a schematic diagram to illustrate USB-PO better. Please refer to point 8 in global response .**
>
> ___
>
> ## 2. Elaborate the algorithm that has heavy dependence on the threshold
> CMLO sets a threshold for model shift to satisfy the monotonic performance improvement guarantee. We argue that this threshold plays a determinant role in the whole algorithm.  ‘’Theorem 4.6 (Refined Bound with Constraint)’’ in CMLO stated
> $$
> V^{\pi_2|M_2}-V^{\pi_1|M_1} \ge \kappa (\mathbb{E}\_{s,a\sim d^{\pi_1}}D\_{TV}[P(\cdot|s,a)||P_{M_1}(\cdot|s,a)] - \mathbb{E}\_{s,a\sim d^{\pi_2}}D\_{TV}[P(\cdot|s,a)||P_{M_2}(\cdot|s,a)])   - \frac{\gamma}{1-\gamma}L(2\sigma_{M_1,M_2}) - \epsilon_{opt}
> $$
> , where $\sigma_{M_1,M_2}$ denotes the threshold to constrain model shift. Obviously, **to meet the monotonic performance improvement guarantee, the right-hand side of the upper inequality must be greater than 0**. Thus, this threshold needs to be carefully designed, as we stated in **line 33-37**. To further verify our description, we **devise an experiment** that sets three different thresholds for CMLO on Walker2d environment (1.0 for the lower threshold, 3.0 for the paper recommended and 5.0 for the higher threshold). As Figure 3 in global response pdf shows, the performance corresponding to the other two thresholds (1.0 and 5.0) is severely affected.
>
> ___
>
> ## 3. Trust-Region like updates
> Nice view! We've thought about this idea. **However, this would introduce a new problem in determining the weights to balance the MLE updates (model bias) and trust region constraints (model shift), falling into the same situation as CMLO**. Though our method seems to be a little more complicated, it is **straightforward and effective** under our theoretical framework. Please see **Appendix C.4**, compared to MBPO, USB-PO does not introduce excessive computational resource cost. However, we believe that this idea may work to further optimize USB-PO with some technical support.
>
> ___
>
> ## 4. Related works
> Thank you again for pointing out our omission regarding the relevant literature! According to your advice, we read the relevant papers covering **CPI [21], DPI [41], GPS with unknown dynamics [26] and CDPO [51]** then we update our related works as follows (changes are bolded).
>
> Performance improvement guarantee is a core concern in both MFRL and MBRL theoretic avenues. In MFRL, methods such as TRPO [39] **and CPI [21]** choose to optimize the performance difference bound, whilst most of the previous work in MBRL [29, 19, 50, 36, 23] choose to optimize the difference of expected return under the model and that of the real environment, which is termed return discrepancy. However, return discrepancy ignores model shift between two consecutive iterations compared to the performance difference bound under the MBRL setting, which can lead to performance deterioration due to excessive model updates. Although some recent methods have also employed performance difference bound to construct theoretical proofs, they still suffer from certain limitations. OPC [11] designs an algorithm to optimize on-policy model error, but it is similar to return discrepancy in nature. **DPI [41] uses dual updates to improve sample efficiency but tries to restrict policy updates within the trust region, thus inhibiting exploration.** CMLO [20] relies on a fixed threshold to constrain the impacts of model shift, resulting in a heavy dependence on the threshold and a lack of adaptability during the training process. Hence, we try to unify model shift and model bias to form a novel optimization problem, adaptively fine-tuning the model updates to get a performance improvement guarantee. **Still, some prior work [7, 34, 51] choose to consider regret bound, among which [51] also reduce the impacts of the model changing dramatically between successive iterations. Instead of unifying model shift and model bias, they choose to realize dual optimization by considering maximizing the expectation of the model value rather than that of the single model as a sub-process. Different from [51, 41, 26] that use dual optimization to train the policy, we devise an extra phase to fine-tune the model.**
>
> [21]   Sham Kakade et al. “Approximately optimal approximate reinforcement learning”. In: Proceedings of the Nineteenth International Conference on Machine Learning. 2002, 358
> pp. 267–274.
>
> [41] Wen Sun et al. “Dual policy iteration”. In: Advances in Neural Information Processing Systems 31 (2018).
>
> [26] Sergey Levine et al. “Learning neural network policies with guided policy search under unknown dynamics”. In: Advances in neural information processing systems 27 (2014).
>
> [51] Shenao Zhang. “Conservative Dual Policy Optimization for Efficient Model-Based Reinforcement Learning”. In: Advances in Neural Information Processing Systems 35 (2022), pp. 25450–25463.
>
> ___
>
> We hope the reviewer can consider raising the score if we resolved the reviewer’s concerns and we would be happy to have further discussion.

---

> > ### Comment · Reviewer_C4Te · 2023-08-16
> >
> > Thanks for the authors' effort during rebuttal. Most of my concerns regarding the notations and novelty, especially the connections with previous works are addressed. I have therefore raised my score.

---

> > > ### Author Response · Authors · 2023-08-17
> > > **Thank you for the inspiring reply!**
> > >
> > > Dear reviewer,
> > >
> > > Thank you for helping us improve the paper and update the score! We really appreciate your valuable comments!
> > >
> > > Best wishes!
> > >
> > > The authors.

---

### Author Rebuttal · Authors · 2023-08-09

We are very grateful to all the reviewers for the valuable feedback, helping us improve the paper further!

We revise the paper and add suggested experiments according to the reviewer’s comments. The detailed revisions are as follows. The additional figures and a table are in the pdf file.

___

## 1. Revisions of related work
We revise the line 75-87: Due to the character limit, we put the revision of the related work in the reply to the reviewer C4Te.

___

## 2. Revisions of the statement of Theorem 1
We revise the line 133-135 as: Obviously, compared to directly optimizing the return discrepancy of each iteration **[19]**, the performance difference bound chooses to optimize the return discrepancy of two adjacent iterations, **namely** $V^{\pi_2|M_2}-V^{\pi_2}\_{M_2}$ **and** $V^{\pi_1|M_1}-V^{\pi_1}\_{M_1}$ **respectively**, and the expected return variation between these two iterations, **namely** $V^{\pi_2}\_{M_2} - V^{\pi_1}\_{M_1}$, demonstrating better rigorousness.

[19] Michael Janner et al. “When to trust your model: Model-based policy optimization”. In: Advances in neural information processing systems 32 (2019).

___

## 3. Revisions of Eq.(7) in Theorem 3
We revise the typo as below:

$V^{\pi_2|M_2} - V^{\pi_1|M_1} \ge \frac{2R\_{max}\gamma}{(1-\gamma)^2}(\epsilon^{\pi_1}\_{M_1} - \epsilon^{\pi_2}\_{M_2} - \epsilon^{M_2}\_{M_1}) - \frac{2R_{max}\epsilon\_\pi}{(1-\gamma)^2}$

___

## 4. Revisions of Section 4.1
We revise the line 114-122 as: The general algorithmic framework of USB-PO is depicted in Algorithm 1, **where the main difference compared to the existing MBRL algorithms is the two-phase model learning process, namely phase 1 and phase 2. Phase 1 uses traditional MLE loss to train the model, which may impair the performance by excessive model updates due to only considering the impacts of model bias. To mitigate this problem, we introduce phase 2 to further fine-tune the model updates, whose optimization objective is defined as Eq.(3)**. Eq.(3) unifies the model shift term and the model bias term in the second-order Wasserstein distance form, namely $W_2(p_{M_1},p_{M_2})$ and $W_2(p_{M_2},p_{M^*})$, thus achieving adaptive adjustment of their impacts during the fine-tuning process. As demonstrated in Section 4.2 and Section 5.3, this is not equivalent to the traditional methods of limiting the magnitude of model updates, but rather beneficial to get a performance improvement guarantee.

___

## 5. Revisions of Algorithm pseudo-code
To avoid confusion of notation between $p_M$ and $p_{M_1},p_{M_2}$, we integrate Algorithm 2 into Algorithm 1 and remove Algorithm 2 from the main paper:

| *Algorithm 1* Meta-Algorithm of the USB-PO Framework |
| :----|
| 1: Initialize the policy $\pi$ and the learned model |
| 2: Initialize the environment replay buffer $\mathcal{D}$ and the model replay buffer $\mathcal{D}_M$ |
| 3: **for** each epoch **do** |
| 4: $\qquad$ Use $\pi$ to interact with the real environment: $\mathcal{D}\leftarrow  \mathcal{D}\cup\{(s,a,r,s')\}$ |
| 5: $\qquad$ Backup the current learned model for future use and denote this backed-up model as $p_{M_1}$ |
| 6: $\qquad$ Phase 1: use $\mathcal{D}$ to train the learned model with the supervision of MLE and denote this updated model as $p_{M_2}$ |
| 7: $\qquad$ Phase 2: use Eq.(3) as optimization objective to further fine-tune $p_{M_2}$ |
| 8: $\qquad$ Use $p_{M_2}$ to generate the imaginary rollouts: $\mathcal{D}_M\leftarrow  \mathcal{D}_M\cup\{(s_M,a_M,r_M,s_M')\}$ |
| 9: $\qquad$ Use $\mathcal{D}\cup\mathcal{D}_M$ to train the policy $\pi$ |
| 10: **end for** |

___

## 6. Revisions of the statement of Eq.(3)
To prevent misunderstanding, we revise the line 171-173 as: **Hence, Eq.(3) can be applied as the optimization objective to get a performance improvement guarantee.**

___


## 7. Revisions of time cost
We add the statement of time cost after line 225: **Computational Cost.
We report our computational cost compared to MBPO in Appendix C.4. Although USB-PO is a two-phase model training process, continuing to use the fine-tuned model for the next iteration has the potential to accelerate model convergence and then possibly reduce the training time.**

___

## 8. An added simple schematic diagram
To better illustrate USB-PO, we add an additional schematic diagram. Please see Figure 1 in global response pdf. We will put this into the appendix later.

___

The above is a summary of the revisions. For questions raised by each reviewer, please refer to the rebuttal to the specific reviewer.

---

### Decision · Program_Chairs · 2023-09-21

**Decision:**

Accept (poster)

**Comment:**

All the reviewers were positive about this paper, except Reviewer VRuX, who found the paper to be presented not very clearly. The AC strongly recommends the authors to take their comments into account while revising the paper, but leans towards acceptance due to the positive response to the contribution made in the paper.

That said, I do personally think that the paper is quite lacking in its evaluation -- while the authors in response to the reviewers and the AC argue that DMC tasks are of no need, I would like to note that experiments on Gym tasks with small improvements were an accepted paradigm in 2018-2019 (when the SAC and MBPO papers came out), and since then several RL papers study a broader spectrum of tasks. In the context of the method prpposed in this paper, I believe that the same approach could be instantiated with Dreamer or image-based MBPO, such as with SLAC, and at least some experiments should be added to the camera-ready version of the paper.

With regards to the point made by the authors, I agree that while the Gym domains are sufficient to validate the theory, I think viewing this paper as a theory paper primarily puts it below the acceptance threshold. In contrast, if we consider this as an empirical paper, then empirical contributions in and off themselves should be strong enough, which seems to not be the case in the submission.